# Spatiotemporal refinement of signal flow through association cortex during learning

Ariel Gilad[1,3] & Fritjof Helmchen [1,2✉]

Association areas in neocortex encode novel stimulus-outcome relationships, but the principles of their engagement during task learning remain elusive. Using chronic wide-field calcium imaging, we reveal two phases of spatiotemporal refinement of layer 2/3 cortical activity in mice learning whisker-based texture discrimination in the dark. Even before mice reach learning threshold, association cortex—including rostro-lateral (RL), posteromedial (PM), and retrosplenial dorsal (RD) areas—is generally suppressed early during trials (between auditory start cue and whisker-texture touch). As learning proceeds, a spatiotemporal activation sequence builds up, spreading from auditory areas to RL immediately before texture touch (whereas PM and RD remain suppressed) and continuing into barrel cortex, which eventually efficiently discriminates between textures. Additional correlation analysis substantiates this diverging learning-related refinement within association cortex. Our results indicate that a pre-learning phase of general suppression in association cortex precedes a learning-related phase of task-specific signal flow enhancement.

[1] Brain Research Institute, University of Zurich, CH-8057 Zurich, Switzerland. [2] Neuroscience Center Zurich, CH-8057 Zurich, Switzerland. [3] Present address: Department of Medical Neurobiology, Institute for Medical Research Israel Canada, Faculty of Medicine, The Hebrew University, 9112001 Jerusalem, Israel. ✉email: helmchen@hifo.uzh.ch

The neocortex dynamically changes when we learn new tasks. Learning to discriminate between different stimuli, e.g. visual stimuli or texture touches, leads to changes in the respective primary sensory areas, i.e., primary visual cortex (V1) and barrel cortex (BC)[1–9]. Specifically, experts display enhanced neural responses and discrimination power in these areas compared to naïve subjects[2,7,10,11]. What are the cortical processes—perhaps even before an animal gains expertise—that establish such enhanced stimulus discrimination? Higher-order areas, e.g., retrosplenial cortex and secondary motor cortex, mediate learning-induced cortical modulation via top-down effects[4,12,13] but how spatiotemporal cortical dynamics reorganizes during learning of a specific task remains elusive. Three relevant dimensions to be considered are (1) the large-scale spatial dimension across multiple cortical areas, (2) the time course of individual trials (lasting few seconds), and (3) the learning time course spanning the entire training (hundreds of trials across several days).

With respect to spatial dimension, we previously measured large-scale cortical dynamics with wide-field calcium imaging in mice trained to discriminate two texture types with their whiskers in a go/no-go task[11]. Expert mice displayed enhanced activity for the rewarded go-texture in BC, secondary somatosensory cortex (S2), and rostro-lateral cortex (RL). RL is part of the posterior parietal cortex (PPC)[14,15] within the cluster of higher-order association areas surrounding V1 that play pivotal roles in cross-modal sensory integration[16–20], formation of stimulus-outcome sequences and maintenance of history-dependent information[21–24]. However, we know little about the involvement and interactions of these areas during learning.

The second dimension relates to the event sequence during individual trials. Whereas most studies focus on the time period when the relevant, to-be-learned, stimulus is presented[2–7,13], cortical dynamics before the stimulus is less well characterized. Perhaps in some association areas, after a trial-start cue, enhanced and anticipatory activity develops just before a task-relevant stimulus arrives[2]. Regarding the third dimension of learning progression, most studies either only compare expert to naïve mice[1–3,5,13,25] (i.e. two time points) or sample learning daily (i.e. 3–8 time points[4,6,7]; but see[9,26]). Such low sampling frequency precludes resolving the trial-by-trial development of learning, which in some animals is rather rapid. For example, some cortical areas may display changes before learning takes place whereas other areas might change when task performance actually improves[9,10,26].

Here, we study spatiotemporal cortical dynamics during learning by performing wide-field calcium imaging across neocortex in mice learning a whisker-based texture discrimination task. We chronically measured trial-by-trial layer 2/3 (L2/3) activity in 25 cortical regions during the entire training period of several days. We find learning-related cortical changes—especially in posterior association areas—that we divide into two phases: First, a pre-learning phase, showing suppression in several association areas, followed secondly by an enhancement of a specific task-related cortical activation sequence that emerges in parallel to increasing task proficiency.

## Results

**Texture discrimination learning.** To study learning-related changes in both brain activity and behavior, we trained transgenic mice expressing GCaMP6f in L2/3 excitatory neurons in a head-fixed, whisker-based go/no-go texture discrimination task[27] (Fig. 1a; Methods). We trained five mice to lick upon whisker-touch with a coarse surface texture (P100 sandpaper) and two mice to lick for a smooth P1200 texture. The respective other sandpaper type served as no-go stimulus. In 'hit' trials mice were rewarded for correctly licking for the go texture. They were punished with white noise for incorrectly licking for the no-go texture ('false alarm' trials, FA) and neither rewarded nor punished when they withheld licking for the go and no-go textures ('Miss' and 'correct-rejection', CR, trials, respectively). As mice learned to discriminate between the two textures, we measured large-scale neocortical L2/3 activity in the hemisphere contralateral to stimulation using wide-field calcium imaging through the intact skull[11,28], along with concurrent video monitoring of whisking and body movements (Methods). In total, we imaged seven mice for 5–11 days (2274 to 5626 trials per mouse).

We considered the three spatiotemporal dimensions mentioned above (Fig. 1b). For analysis of the spatial dimension across the entire dorsal cortex, we functionally mapped sensory areas for each mouse during anesthesia. Based on these maps (and skull coordinates) we registered all images to the 2D top-view Allen reference atlas[29] and defined 25 areas of interest, consolidated in four major groups (Fig. 1c, d and Supplementary Fig. 1 with a list of region abbreviations; Methods). We then analyzed signals on the relevant fast time scale of individual trials (Fig. 1e), with a special focus on the period before texture touch as learning-related changes may also be expected early in trial time. We defined three salient time windows: the 'cue-period' (0.1–0.6 s after the stimulus cue) to capture responses to the initial trial-start cue; the 'pre-period' when the texture approaches the whiskers (−1 to −0.5 s relative to the texture stop; mainly before the first whisker-texture touch); and the 'stim-period' during texture touch (−0.5 to 0.5 s relative to texture stop). We did not further analyze the response period because mice licked and moved rigorously during this period, causing widespread cortical activity difficult to interpret (there was no delay period and mice were free to lick outside the response window). The second relevant temporal dimension is the slow time course of learning across days. All mice increased performance with training (5–11 days; ~500 trials/day) and eventually reached high discrimination levels, quantified by d-prime (d′) values in 50-trial bins (Fig. 1f; refs. [11,27]; Methods). Performance improved mainly due to increased CR rate (Supplementary Fig. 2). For each mouse, we defined the 'learning threshold' for reaching expert level as the crossing point of the learning curve at d′ = 1.5 (in units of 'trial number'), and the 'naïve' and 'expert' phases as the first and last 500 trials with imaging, respectively. The fastest-learning mouse reached threshold in slightly less than thousand trials whereas mouse #7 took substantially longer (Fig. 1g). Some mice displayed a steep learning curve whereas others showed gradual learning, probably reflecting natural variability across mouse individuals. Jointly, these definitions of cortical areas, trial periods of interest, and naïve-to-expert learning phases enabled us to reveal key learning-related changes in both behavior and L2/3 activity across the cortex.

**Motor behavior changes during learning.** We first quantified changes in motor actions during learning. Mice may start moving more when they begin to associate the go-texture with the upcoming reward. Because movements are associated with widespread cortical activity[11,30,31], changes in motor behavior potentially confound the interpretation of learning-related activity changes. Indeed, expert mice indicated their future action before the response cue, by moving their body and by whisking and licking rigorously. To quantify body movements, we detected forelimb and back movements in the body videos and calculated the movement probability across trial time and across learning (50-trial bins; ref. [11]; Methods). When mice approach their learning threshold, they start moving their body clearly before the response cue and reward consumption, resulting in a significantly

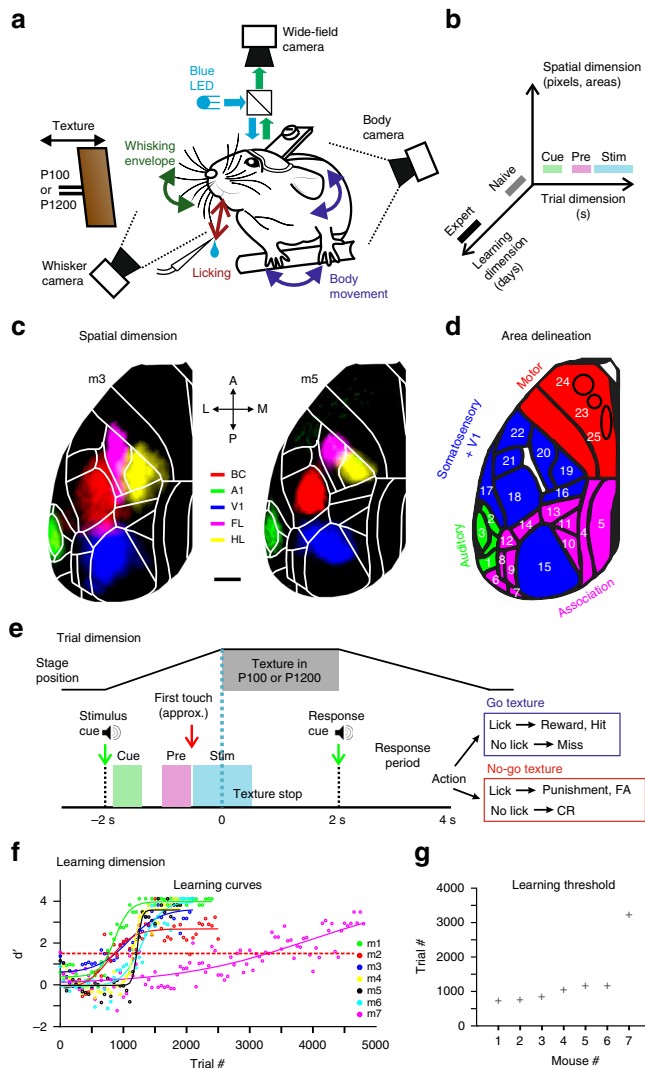

**Fig. 1 Spatiotemporal dimensions relevant for texture discrimination learning. a** Behavioral setup. **b** Schematic of the three relevant dimensions. **c** Functional maps for two example mice (m3 and m5) obtained by overlaying sensory-evoked activity maps for different sensory stimuli. Maps are registered to the Allen reference atlas (white outlines; ©2004 Allen Institute for Brain Science. Allen Mouse Brain Atlas. Available from: http://mouse. brain-map.org/). Scale bar is 1 mm. **d** Area definitions used in this study (see also Supplementary Fig. 1). Four rough divisions are auditory areas (green), association areas (pink), somatosensory + V1 areas (blue), and motor areas (red). **e** Trial structure and possible trial outcomes. After an initial auditory tone ('stimulus cue'), the P100 or P1200 sandpaper approached and stayed in place for 2 s. At withdrawal start, an additional auditory tone ('response cue') signaled the beginning of the report period. Cue-, pre-, and stim-period as analysis windows are marked by different colors. **f** Performance (d′) for all mice across the entire learning period, fitted with a sigmoid function. Red dashed horizontal line indicates threshold for learning (d′ = 1.5). **g** Learning threshold for all mice in ascending order.

higher movement probability in the stim-period for expert vs. naïve phase (Fig. 2a, b and Supplementary Fig. 3; *p* < 0.05, *n* = 7 mice, Wilcoxon signed-rank test). In the pre-period, movement probability was relatively low with insignificant change from naïve to expert (*p* = 0.47), whereas in the cue-period movement probability showed a significant reduction in expert mice from a relatively low naïve level (*p* < 0.05; Wilcoxon signed-rank test).

To understand how body movements relate to learning, we calculated the average movement probability during the stim-period for each mouse throughout learning (Fig. 2c). With a movement threshold of 0.3 (i.e. moving in 30% of trials) we could accurately predict the learning threshold (Fig. 2d; *r* = 0.99, *p* < 0.001 comparing learning and movement thresholds across mice). Thus, although not yet receiving any reward during the stim-period, mice started to move more extensively shortly after texture sensation almost exactly at the time during training when their discrimination performance improved.

Similarly, we analyzed whisking and licking behavior. On average, whisking amplitude significantly increased with learning in the stim-period (*p* < 0.05, *n* = 7 mice, Wilcoxon signed-rank test) but changed little in cue- and pre-period (Fig. 2e, f). Increased whisking during and following texture touch occurred in parallel to the learning curve, similar to body movements, so that the learning threshold could be well predicted with a whisking threshold (Fig. 2g, h). Licking during go-trials was reduced in expert animals in the cue- and pre-period—highlighting the key requirement of lick suppression for learning—whereas it was enhanced in the stim-period (Fig. 2k, l). The increase was not significant, though, presumably because pronounced licking started only at the stim-period end but clearly was enhanced thereafter in expert mice. We conclude that mice exhibit consistent learning-related changes in motor behaviors, engaging their body to solve the discrimination task and collect reward. Once mice learned to discriminate between textures, they initiate various movements during the stim-period whereas they remain relatively quiet before texture touch (in cue- and pre-period). Focusing on the early trial-time periods before touch allowed us to study learning-related changes of cortical processes that lead up to the task-relevant stimulus without the confound of movement-related cortical activity.

**Learning-related changes in cortical activity**. We next analyzed spatiotemporal dynamics of L2/3 cortical activity across learning, as revealed by wide-field calcium imaging. Here, we mainly present results for go-trials (hit and miss trials pooled together). We calculated activity maps by averaging ΔF/F signals during the cue-, pre-, and stim-period, respectively, and compared maps for naïve and expert phase (first and last 500 trials). Maps from two example mice show activation during the cue-period in A1 (also anterior-medial and hindlimb areas) and lower activation in postero-medial (PM), and retrosplenial-dorsal (RD) association areas (Fig. 3a). During the pre-period, RL displayed high activation especially in expert mice. During the stim-period, BC displayed the highest activation level. These example maps indicate a specific sequence of activity emerging during learning, starting from auditory cortex in response to the stimulus cue, followed by RL activation as the texture approaches the whiskers, and continuing to BC activation during touch sensation. This notion of sequential activation is further corroborated by plotting the time course of mean ΔF/F traces in A1, RL, and BC in experts (Fig. 3b for the two example mice; Supplementary Fig. 4a for each mouse separately; Supplementary Fig. 4b for average responses across mice). We also found, especially in experts, that BC displayed enhanced activity prior to the stim-period, initiating just after the stimulus cue (Fig. 3b). This finding implies anticipatory activity in BC that develops during learning. To quantify anticipatory responses in BC, we aligned the BC trace for each trial to first-touch time and computed first-touch-triggered responses by averaging. BC displayed an initial mild rise in pre-touch activity followed by a salient response to texture touch (Supplementary Fig. 5a). The onset of this initial rise occurred significantly earlier in expert compared to naïve mice (*p* < 0.05; Wilcoxon signed-rank test; Supplementary Fig. 5b). This early onset could

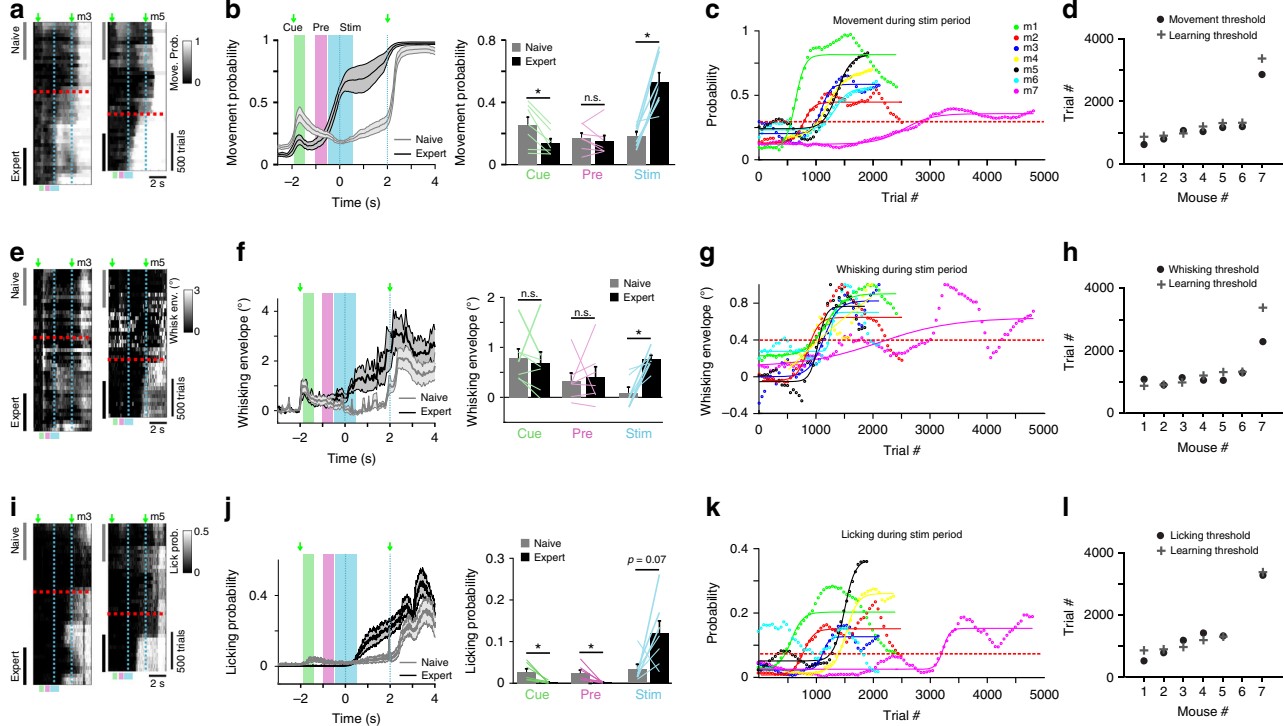

**Fig. 2 Motor parameters during the stim-period are associated with learning. a** Movement probability for go-trials of two example mice plotted as heat maps along the two temporal dimensions (trial dimension on *x*-axis; learning dimension on *y*-axis; 50-trial bins along learning dimension). Horizontal red dashed line indicates learning threshold. Vertical cyan dashed lines demarcate 'texture in' period. Naïve and expert phases were analyzed in the 500 first and last trials, respectively. **b** Left: Time course of movement probability averaged across all 7 mice during go-trials for naïve phase (gray trace) and expert phase (black trace), respectively. Error shading indicates s.e.m. Green arrows indicate stimulus and response cue. Right: Mean movement probability (±s.e.m) in cue-, pre-, and stim-period for naïve and expert phase. *n* = 7 mice; Lines depict individual mice. **p* < 0.05, n.s. not significant, Wilcoxon signed-rank test. **c** Movement probability during stim-period across learning for all mice. Each curve was fitted with a sigmoid function. Dashed red line indicates movement threshold (=0.3). **d** Movement threshold for all mice (circles). For comparison the learning thresholds from Fig. 1g are shown (crosses). **e–h** Same plots as a-d for whisking behavior. The envelope amplitude of whisking was analyzed. Whisking threshold was defined as 0.4°. **i–l** Same plots as a–d for licking behavior. The probability of licking was analyzed. Licking threshold was defined as 0.075. Licking threshold could not be detected for mouse #6.

represent anticipatory activity, but possibly also different motor parameters. To dissociate between anticipatory effects and motor parameters we aligned the body movement vector to first touch. The onset of BC responses occurred significantly earlier than the movement onset (*p* < 0.05; Wilcoxon signed-rank test; Supplementary Fig. 5c, d; results were similar for whisking envelope), implying that mice display a plastic expectation signal in BC that develops during learning.

Next, we scrutinized how the cortical activation sequence from cue- to pre- to stim-period changes across learning. Surprisingly, we found learning-related changes during these early trial periods before texture touch. Based on the example activation maps we focused on 5 areas of interest during specific time periods: A1, PM, and RD during cue-period; RL during pre-period; and BC during stim-period. For each area, we plotted the heat map of trial-related ΔF/F signals across learning, naïve and expert average ΔF/F traces, and the mean ΔF/F responses in the respective trial period across learning (Fig. 3c). In the cue-period, A1 activity displayed variable changes during learning, increasing in one mouse while slowly decreasing in the other (see below). Interestingly, RD and PM showed a suppression of responses across learning for the cue-period. In contrast, RL responses in the pre-period and BC activity in the stim-period consistently increased during learning. These examples show that large-scale cortical activity varies between mice. Some of this variability can be explained by differences in movement parameters. For example, mouse #6 in the expert phase whisked more during the

stim-period compared to mouse #3 (Supplementary Fig. 3b). This may explain its enhanced activity in frontal whisker motor cortex (Fig. 3a). Other differences, for example variable A1 responses during the cue-period, cannot be explained by simple motor parameters and may reflect intrinsic differences between mice. Thus, cortical areas display diverse learning-related changes during specific trial periods, ranging from enhancement (e.g., RL and BC) to suppression (e.g., PM and RD).

We expanded our analysis to all 25 cortical areas by calculating the mean ΔF/F activation for each area during the cue-, pre-, and stim-period and averaging across all mice for expert and naïve phase (Fig. 3d). During the cue-period, several association areas, including PM and RD, showed reduced activation in expert mice. During the pre-period, RL showed the strongest activation in experts whereas PM and RD maintained lower activation levels. Notably, RL still displayed strong activation during the pre-period when we positioned the texture out of reach of the whiskers in two expert mice, thus omitting the relevant stimulus (Supplementary Fig. 6). Thus, RL responses appear not to directly relate to texture touch per se and possibly rather represent the expectation of an upcoming touch. Finally, BC displayed the highest stim-period activation in expert mice, along with activation of other sensory and motor areas, presumably reflecting initiation of movements during this period.

**Wide-spread suppression followed by specific enhancement.** Imaging cortical activity longitudinally throughout the entire

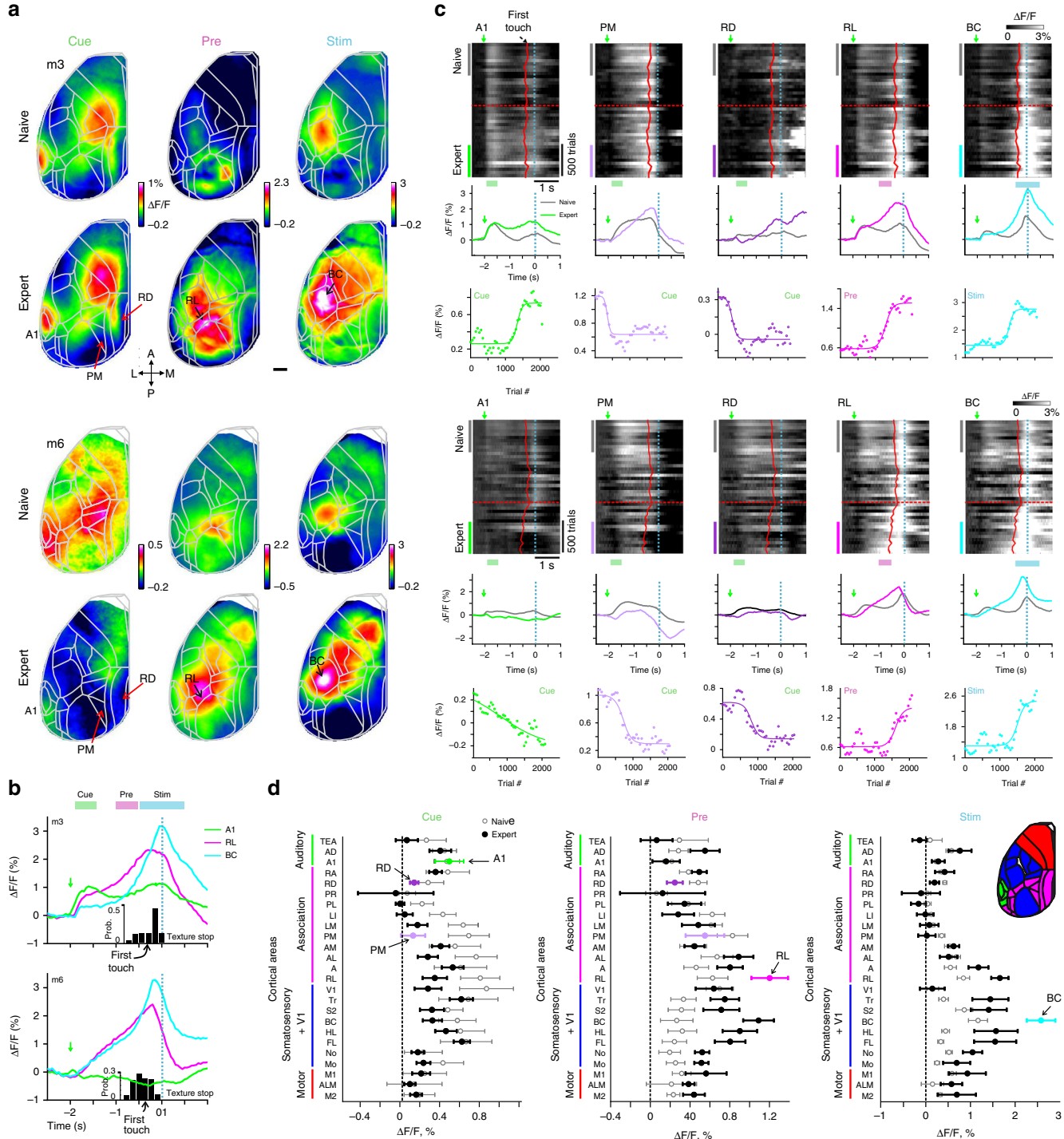

**Fig. 3 Changes in wide-field calcium signals across cortex during learning. a** Example activation maps from two mice averaged during cue-, pre-, and stim-period in naïve (top) and expert (bottom) phase. Color scale bar indicates min/max of percent ΔF/F. Overlay of areas in gray for all maps. 5 areas of interest are marked. Scale bar is 1 mm. **b** Expert trial-related average responses in A1, RL, and BC for the two example expert mice. Green arrow marks stimulus cue. Histograms at the bottom indicate distribution of the first touch of the whiskers on the texture. **c** Continuation of examples in **a**. Top: population responses plotted for the two temporal scales (trial scale along x-axis; learning scale along y-axis) for five areas of interest (from left to right): A1, PM, RD, RL, and BC. Data is binned every 50 trials. Red lines indicate mean first touch of the whiskers on the incoming texture. Dashed cyan line indicates texture stop. Dashed red line indicates learning threshold. Middle: trial-related area responses for expert (colored) and naïve (gray). Bottom: mean responses across learning for each area averaged during the specific trial period indicated. Each curve is fitted with a sigmoid function. **d** Mean activation of all 25 cortical areas in expert (black) and naïve (gray) mice during cue-, pre-, and stim-period and grouped into auditory (green), association (pink), somatosensory + V1 (blue), and motor (red) areas (see also inset). Error bars are s.e.m. across mice ($n = 7$).

learning process enabled us to relate the time course of regional learning-related activity changes more precisely to the behavioral learning curve. First, we applied sigmoidal fits to the mean $\Delta F/F$ changes for our areas of interest and normalized the curve fits to compare time courses (Fig. 4a; see Supplementary Fig. 7 for non-normalized traces for all mice). Surprisingly, PM and RD showed suppression long before the enhancement in BC and RL and even clearly before the behavioral learning threshold was reached. The inflection point (the point of maximal steepness; for normalized curves at 0.5-crossing) occurred significantly earlier for PM and RD than for RL and BC and also significantly preceded the learning threshold ($p < 0.05$; Wilcoxon signed-rank test; Fig. 4b individual mice; Fig. 4c average across mice). PM and RD suppression occurred about 500 trials before the mouse reached learning threshold and before RL and BC enhancement. In addition, for each of these four areas the inflection point positively correlated with the learning threshold across mice ($r = 0.97$, 0.97, 0.79, and 0.79 for BC, RL, PM, and RD, respectively; $p < 0.05$). Consequently, the early suppression in PM and RD could predict well when the mouse will learn the task, hundreds of trials in advance. In contrast, the inflection points for BC and RL were not significantly different from learning thresholds ($p > 0.05$; Wilcoxon signed-rank test), implying that these changes occur rather in parallel to increases in d′ and thus cannot predict when learning threshold is reached.

A closer look at the variability of learning-related changes of A1 cue-period activity (Fig. 3c) revealed that suppression and enhancement were discernible as two consecutive phases in the A1 signals (Fig. 4d and Supplementary Fig. 8). Suppression consistently occurred before mice reached learning threshold whereas enhancement occurred thereafter. The relative amplitude of modulations (suppression or enhancement) varied between mice but cue-period calcium signals were significantly lower in amplitude in A1 around the time of learning compared to naïve and expert phases (Fig. 4e; $p < 0.05$; Wilcoxon signed-rank test; averaged across ±100 trials

around threshold). Thus, learning-related activity changes are not necessarily uni-directional, i.e. exclusively decreasing or increasing, they may display mixed effects. Apparently, cue-induced A1 activity is suppressed early during training before learning, similar to PM and RD, but dynamically shifts to enhancement after the learning threshold has been reached, similar to RL and BC, albeit to a variable degree. We wondered whether such two phases of pre-learning suppression and learning-related enhancement also exist in other cortical areas (and for different trial periods) and fitted the learning-related $\Delta F/F$ signals in all areas with a two-phase sigmoidal model in cue-, pre- and stim-period. The results corroborated the concept of two phases with substantial pre-learning suppression in association areas and later specific enhancement of activity in task-relevant areas in congruence with learning (Supplementary Fig. 9).

As alternative approach to quantify the relationship of task performance and neural activity during learning, we defined 'learning maps' for cue-, pre-, and stim-period by correlating the d′ learning curve with the corresponding time course of $\Delta F/F$ signals for each pixel (averaged over the respective trial period; Fig. 5a). Example learning maps for two mice reveal that several association areas display negative correlation values during the cue-period (Fig. 5b), reflecting predominant suppression in these areas during this early period. During the pre-period, RL displayed the highest positive correlation whereas specifically PM and RD maintained negative values. Finally, during the stim-period many sensory and motor areas showed strong correlation, including BC, presumably reflecting motor-related neural activity. The divergence of activity patterns across areas during the pre-period (positive correlation with learning in RL and BC; low or negative correlations in PM and RD) is obvious when plotting correlations with the learning curve for each acquisition frame during the trial period (Fig. 5c). Pooled across mice, correlations between activity and learning were mostly negative in our 5 prime areas during the cue-period, then became significantly positive in RL and BC for the pre-period (while staying significantly negative for PM and

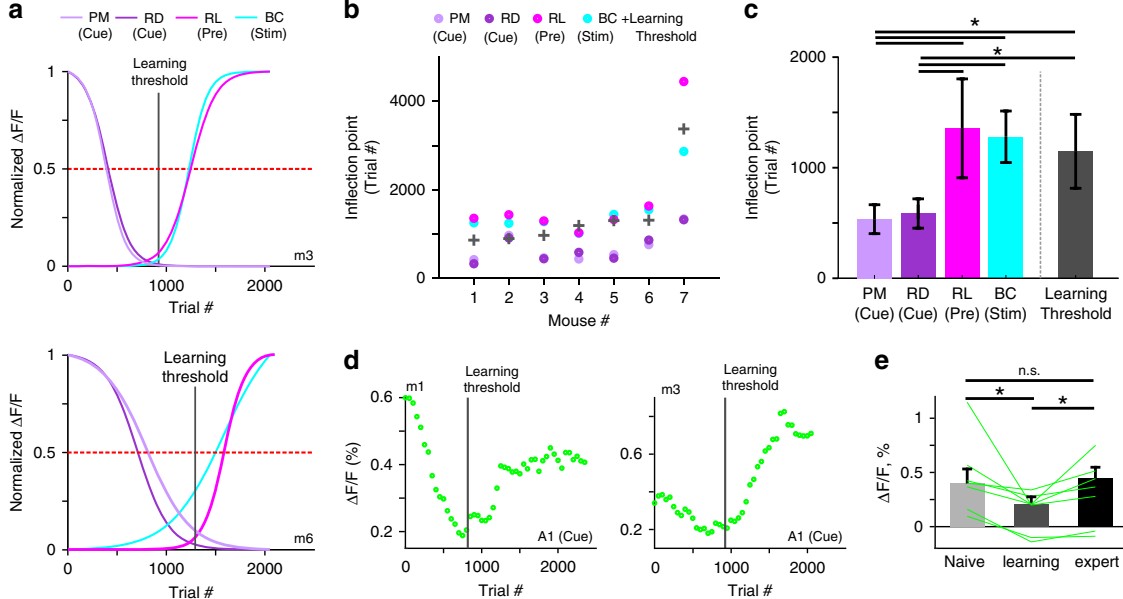

**Fig. 4 PM and RD suppression occurs before learning and precedes RL and BC enhancement. a** Normalized sigmoidal fits to the learning-related mean $\Delta F/F$ changes in PM and RD (for cue-period), in RL (for pre-period), and in BC (for stim-period) for two example mice. Horizontal dashed red line indicates 0.5-level for determining steepest points of change. Vertical solid gray line indicates the learning threshold of the mouse. **b** Steepest points of change for all mice in each of the four areas. Learning threshold is marked with a gray plus sign (similar to Fig. 1g). **c** Steepest points of change (inflection points) and learning threshold averaged across mice. Error bars are s.e.m. across mice. **d** Learning-related $\Delta F/F$ changes in A1 during the cue-period for two example mice. Vertical gray line indicates learning threshold. **e** Mean $\Delta F/F$ changes in A1 during the cue-period for naïve, learning, and expert mice. Error bars are s.e.m. across mice. Lines depict individual mice. *$p < 0.05$; n.s. not significant; Wilcoxon signed-rank test.

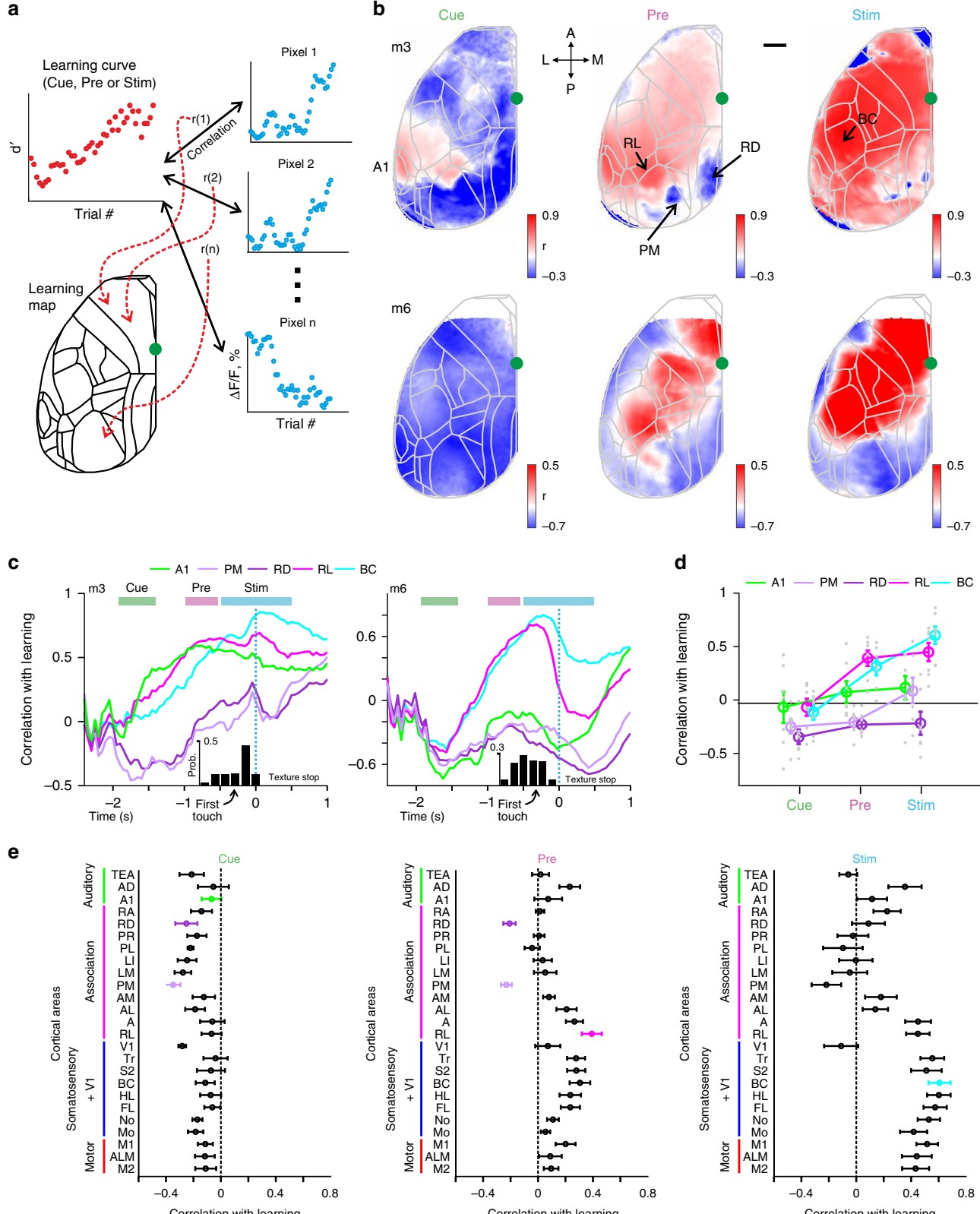

**Fig. 5 Learning maps reveal dissociation of association areas in relationship to learning. a** Schematic illustration for calculating a learning map. Each pixel in the maps reflects the correlation coefficient ($r$) between the mouse's learning curve and the curve of learning-related $\Delta F/F$ change of the respective pixel. The latter can be averaged across the key trial periods (i.e. cue, pre, or stim) or calculated for each time frame. **b** Learning maps during cue-, pre-, and stim-periods in two example mice. Color denotes $r$-values. Scale bar is 1 mm. **c** Correlation with learning as a function of time for the 5 key areas in two example mice. Histograms at the bottom indicate distribution of the first touch. **d** Correlation with learning during cue-, pre-, and stim-periods for the 5 areas. Error bars are s.e.m. across mice. Gray dots depict individual mice. **e** Correlation with learning during cue-, pre-, and stim-periods for all areas. Error bars are s.e.m. across mice.

RD), and significantly positive in BC for the stim-period (along with RL; Fig. 5d; $p < 0.05$; Wilcoxon signed-rank test). Across all 25 areas, many association areas were negatively correlated with d' values during the cue-period, followed by a spatial refinement during the pre-period, with RL displaying positive correlation with learning whereas PM and RD exclusively pertained negative correlations (Fig. 5e). As highlighted by the learning maps, BC as well as most of the sensory and motor areas displayed positive correlation for the stim-period. We conclude that association areas undergo a spatial refinement during learning, especially during the trial period bridging the initial stimulus cue and the arrival of the texture as task-relevant stimulus.

**Dissociation within the association network.** We next further quantified the differences and potential interactions among different areas during learning. Similar to the learning maps we

calculated 'seed maps', for which we correlated the learning-related $\Delta F/F$ changes for all pixels with the reference learning-related $\Delta F/F$ change in a 'seed' area (Fig. 6a). Guided by the learning maps, we first calculated pre-period maps with association areas RL, PM, or RD as seed. The RL seed map revealed a positive correlation of activity in this area with sensory and motor cortices as well as with adjacent association areas (Fig. 6b). In contrast, PM and RD seed maps showed high correlations among each other and with their adjacent areas but lower correlations with RL, BC, and most other cortical areas (Fig. 6c; pooled across mice).

A closer look at the inter-areal correlations in the posterior part of cortex revealed that all association areas were highly correlated during the cue-period but varied largely during the pre-period (Fig. 6d). Based on anatomical projections[29,32] and our functional observations from the seed maps (Fig. 6b), we divided the association cortex into anterior (RL, A, AM, and AL) and posterior (PM, RD, LM, LI, PL, PR, and RA) areas (dashed red

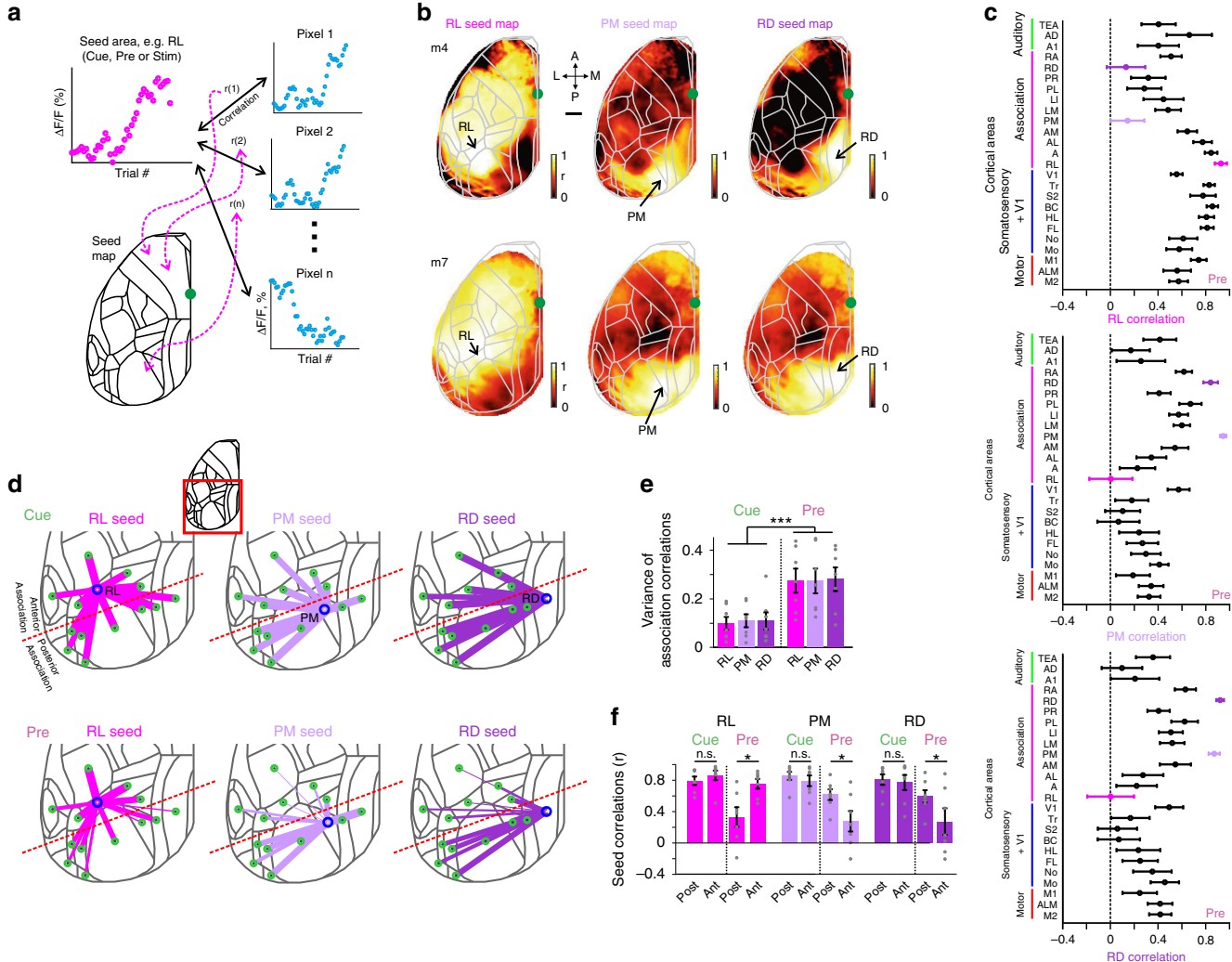

**Fig. 6 Functional reorganization of association cortex during learning. a** Schematic of calculating a seed correlation map. Each pixel in the map reflects the correlation coefficient ($r$) between the learning-related $\Delta F/F$ curve of a seed area (e.g. RL, PM, or RD) and the learning-related $\Delta F/F$ signal changes of the respective pixel. Seed maps can be calculated separately for cue-, pre-, or stim-period. **b** Seed maps for RL, PM, and RD during pre-period in two example mice. Color denotes r-values. Scale bar is 1 mm. **c** Correlation between the three seed areas and all the other areas during pre-period, averaged across all mice. Error bars are s.e.m. across mice. **d** 2D overview of the inter-areal correlations across learning during cue- (top) and pre-period (bottom) between RL (left), PM (middle), and RD (right) and the surrounding cortical areas (also including A1, BC, and V1). Line width is proportional to the average r-value across all mice. Red dashed line indicates separation into anterior and posterior association areas. **e** Variance of inter-areal correlations within all association areas for each seed area in cue- and pre-period. Error bars are s.e.m. across mice. Gray dots depict individual mice. **f** Seed area correlation values for the three areas during cue- and pre-period, averaged separately for anterior or posterior association areas. Error bars s.e.m. across mice. Gray dots depict individual mice. *$P < 0.05$. ***$P < 0.001$. n.s. not significant. Wilcoxon signed-rank test.

line in Fig. 6c). For all three seed areas, the variance of correlation with other association areas was higher during the pre-period compared to cue-period (Fig. 6e; $p < 0.001$; Wilcoxon signed-rank test). Moreover, during the pre-period RL displayed significantly higher correlation with anterior compared to posterior association areas whereas PM and RD displayed the opposite effect (Fig. 6f; $p < 0.05$ for pre-period; $p > 0.05$ for cue-period; Wilcoxon signed-rank test; see full correlation matrix for learning-related $\Delta F/F$ changes in all areas and trial periods in Supplementary Fig. 10). In summary, the network of association areas is reorganized and spatially refined during learning, showing enhanced and correlated activity in anterior areas but perpetual suppression in posterior areas, specifically before texture touch.

**Barrel cortex discriminates best between textures.** So far, we focused on go trials and learning-related changes of large-scale cortical dynamics. Most effects, e.g., widespread suppression followed by specific enhancement involving RL, relate to trial periods before texture touch and thus were also present in no-go trials. However, which are the areas that eventually develop discriminative power to distinguish texture types? We first concentrated on the primary sensory area, i.e. BC[2,7,10,11]. The average time course of trial-related $\Delta F/F$ signals in BC was the same in go and no-go trials in naïve mice (Fig. 7a). In expert mice, in contrast, touch-evoked $\Delta F/F$ changes were generally enhanced in go trials and substantially higher than in no-go trials (Fig. 7a). We computed the go/no-go discrimination power at single-trial level using receiver operating characteristics (ROC) analysis[27,33], with the area under the curve (AUC) relating to discrimination power (Methods). AUC values in BC increased during learning and were significantly higher for the stim-period in expert vs. naïve phase (Fig. 7b, c). Discrimination power increased shortly after the first whisker-to-texture touch (several hundred milliseconds before texture stop; see Supplementary Fig. 5e for AUC aligned to first touch). Similarly, we calculated AUC values during the stim-period for all 25 areas in naïve and expert mice. Pooled across mice, BC displayed the highest AUC value, followed by motor,

sensory and frontal association areas (Fig. 7d). The highest discrimination power in BC is further highlighted when computing AUC values pixelwise and creating an AUC map (Fig. 7e).

Finally, we further investigated stim-period AUC changes in BC across learning (Fig. 7f). In all mice, AUC values increased with learning and the learning threshold of each mouse correlated with the inflection point of the sigmoidal AUC curve fit (Fig. 7f, g; $r = 0.98$; $p < 0.05$). Thus, discrimination power in BC emerges at exactly the time when mice pass the learning threshold, indicating the tight linkage between cortical reorganization and improved task performance. Nevertheless, other areas in somatosensory and motor cortex develop high discrimination power during the stim-period, too, possible relating to motor parameter changes (Fig. 2) and highlighting the large extent of learning-related modulations. We performed additional single-trial analysis by calculating the trial-to-trial variance for naïve and expert mice (first and last 500 go-trials), revealing significantly higher variance in expert compared to naïve mice, mainly in somatosensory cortices and M1 and only during the stim-period ($p < 0.05$; Wilcoxon signed-rank test). A possible explanation is that expert mice increase their body movements both in amplitude and variability during the stim-period (Fig. 2b), which may lead to higher trial-to-trial variability in the respective areas.

## Discussion

We have identified learning-related changes in cortical activity covering a wide range of spatiotemporal dimensions. First, changes were distributed across many cortical areas and comprised suppression, enhancement, and sequential combinations thereof. Second, salient changes occurred in early trial periods before texture-touch, indicating that understanding the trial structure and grasping the context, within which the relevant stimulus is embedded, is an essential part of learning. Third, decreases in cortical activity occurred consistently several hundred trials before actual learning, suggesting that preparatory changes are required before subsequent cortical adaptations actually lead to improved task proficiency. The main pattern we observed is an

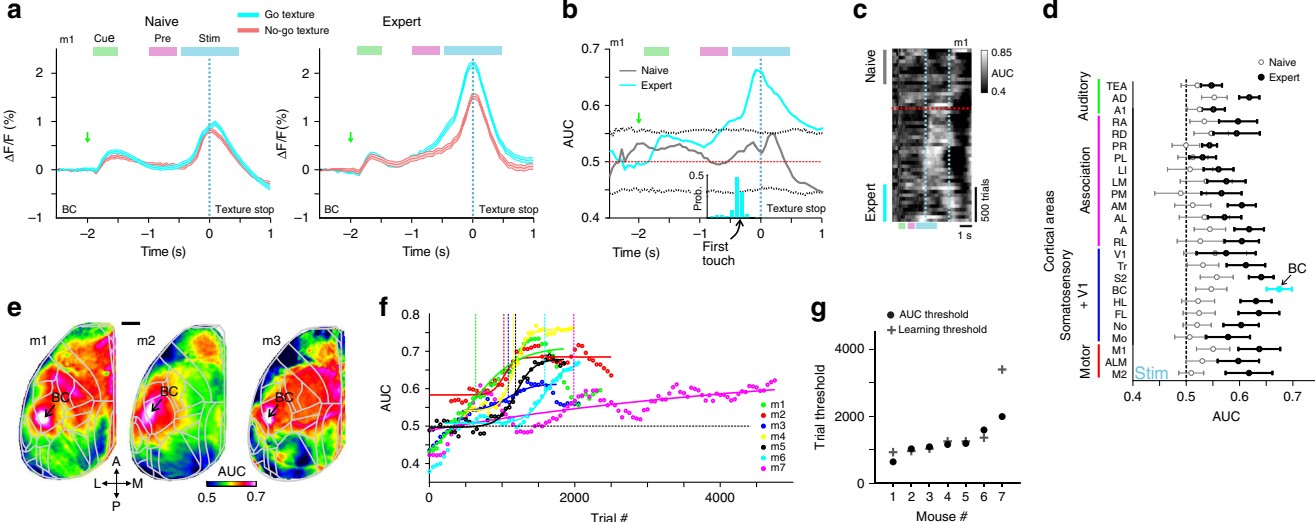

**Fig. 7 Emergence of discrimination power in barrel cortex during learning. a** BC activity in an example mouse for go (cyan) and no-go (red) trials in naïve (left) and expert (right) mouse. Error bars are s.e.m. across trials. **b** ROC-AUC values for go vs. no-go trials as a function of time for naïve (gray) and expert (cyan) mice. Dashed gray lines indicate mean ± 2 s.d. of shuffled data. Same example as in **a**. Histogram below depicts the distribution of the first touch for the expert case in this example. **c** Heat map of trial-related AUC values across learning dimension (vertical axis) in the same example mouse. Dashed red line indicates learning threshold. Dashed cyan lines indicate texture-in period. **d** Pooled AUC values during stim-period in all areas for expert (blue) and naïve (black) mice. Error bars are s.e.m. across mice. **e** AUC maps during stim-period in three example expert mice. Color denotes AUC values. Scale bar is 1 mm. **f** AUC values in BC during stim-period across learning for all mice. Threshold for each mouse is indicated with a vertical line at the inflection point of the sigmoid fit. **g** Inflection points of AUC curves defining 'AUC thresholds'. Learning thresholds are marked with gray plus signs.

early widespread suppression in association areas followed by an enhancement of a spatially more confined set of task-relevant areas (Fig. 8). Eventually, the emergence of a robust trial-related activation sequence from auditory cortex to RL to BC leads to the highest neural discrimination power in BC upon touch.

The activation of RL (as part of PPC) before texture touch may reflect anticipatory, predictive, or attentional processes, as reported previously in primate PPC (area LIP)[34,35]. Accordingly, RL in mice displays predictive responses when the texture stimulus is omitted[36] (Supplementary Fig. 6). In addition, anatomical projections from RL to BC[29,32] imply top-down processing that may aid the preparation of adequate processing of texture information in BC and the association of go-stimulus with future reward. Consistent with this notion, the establishment of a robust temporal sequence, with RL bridging the stimulus-cue to the task-relevant texture stimulus, follows a similar time course as the divergence of BC signals for hit and CR trials. Emerging discrimination power in BC also goes hand in hand with increases in body movements, apparently in expectation of and preparation for upcoming reward. These extensive movements result in widespread and large cortical activity, including forelimb cortex and motor areas. The increasing mix of task-related stimulus processing and behavior-related activation patterns in later trial phases makes it difficult to separate these aspects after texture touch. Further experiments are required to dissect the cortical signal flow for conversion of touch information into preparatory and executive motor signals.

Regarding the temporal dimension of learning across several days, we found a suppression of several association areas, especially PM and RD, as a particularly salient event (Fig. 8). This early suppression was obvious around 500 trials before a mouse learns to discriminate textures, at a time when mean performance was still low ($d' = 0.32 \pm 0.31$ mean ± s.e.m.; most mice still licking for both textures). Because suppression occurred consistently before learning onset, we could even predict from it when the mouse is likely to reach learning threshold. Since we found suppression in other association areas, a common inhibitory control mechanism may be at work. In our interpretation, suppression during the cue-period may indicate a general attentive state as prerequisite for learning. In auditory cortex, we found a combination of early suppression followed by later enhancement, indicating that mice may use the stimulus cue information in order to prepare for the upcoming trial. A1 responses to the stimulus cue did not significantly differ between expert and naïve mice but were clearly reduced during the steepest part of the learning curve (Fig. 4e). This result points to a pronounced reorganization during this training phase that may involve several factors such as inhibitory effects, excitation-inhibition balance, synaptic plasticity or top-down interactions. Whereas here we focused on cortex-wide L2/3 excitatory neurons, learning-related dynamics may involve other circuit elements such as deep cortical layers[37,38], inhibitory subtypes[39] or subcortical areas[38,40]. For example, activity of L6 neurons decreases the gain modulation of superficial layers[37], which could contribute to the initial global suppression in association areas. Another possibility is that higher-order thalamocortical connections may drive synaptic plasticity during learning[38], perhaps specifically in distinct association areas such as RL[40]. These factors may also contribute to the enhanced go/no-go discrimination in BC in expert mice, because a possible integration of anticipatory signals from RL and higher-order thalamocortical inputs may enhance the population activity in BC for the go texture. Future studies should expand to

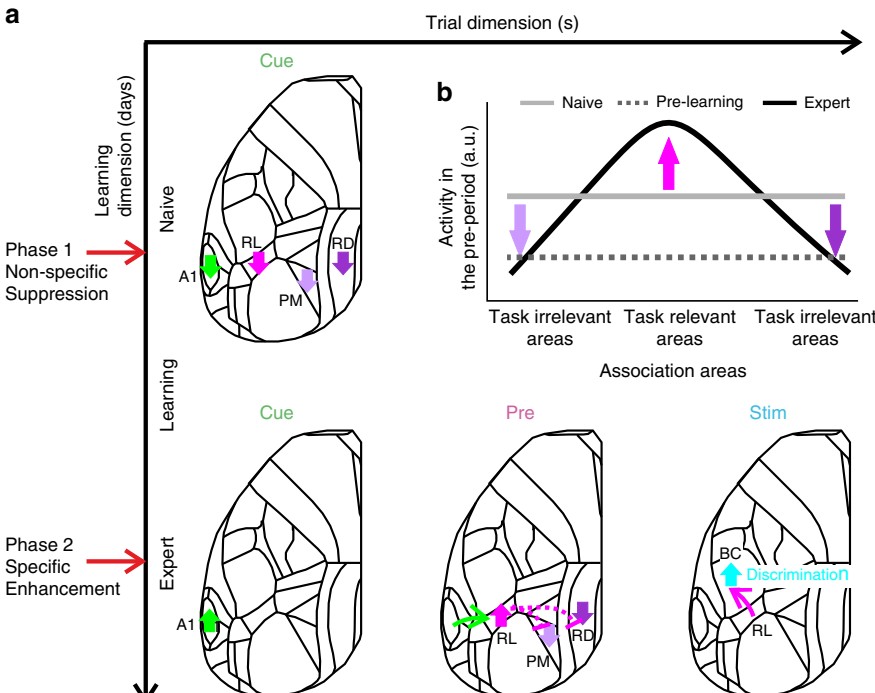

**Fig. 8 Learning starts with a general suppression phase followed by a specific enhancement phase. a** A schematic illustration of the main cortical changes with regard to the two temporal scales: trial (x-axis) and learning (y-axis). Two phases occur across learning: before mice actually learn to discriminate between textures, association areas display non-specific suppression of activity in response to the cue (phase 1: non-specific suppression). Only later, as mice learn the task, a specific sequential pattern is enhanced: starting from A1 in response to the cue, then RL just before the texture touches the whiskers, and BC during texture touch (phase 2: specific enhancement). Association areas PM and RD specifically remain suppressed. **b** Schematic diagram showing the distribution of activity across association areas during the pre-period (i.e. in preparation of the upcoming texture touch) for the naïve (gray), pre-learning (dashed gray), and expert (black) phase.

other cortical layers, cell-types, and brain areas to gain a more comprehensive understanding of learning-related mechanisms.

Our study focuses on wide-field observations across the learning time course, but the causal role of each area during specific time periods remains unknown. Silencing BC during sensation in expert mice impairs discrimination and detection performance (refs. [41–44]; but see Hong et al.[45]). However, are the observed pre-stimulus cortical activity patterns necessary for discrimination learning? RL was found to be causally linked to performance during different discrimination tasks during the stimulus period[16,18,46,47], but little is known about the role of RL during pre-stimulus periods. RL (as part of PPC) is causally linked to maintaining history-dependent information from previous trials[23], suggesting a possible involvement in pre-stimulus periods. As for the initial suppression in PM and RD, we speculate that activating these areas before learning onset (thus equalizing activation across association areas) may prolong the learning curve of the mouse. In contrast, boosting the salient activation pattern across association areas (e.g. enhancing RL while suppressing PM and RD) may facilitate learning. Future studies will focus on the effects of manipulating different association areas during pre-learning periods.

Interestingly, we found dissociation of signaling between anterior and posterior sets of association areas, possibly reflecting distinct roles in neural processing. Anatomical evidence suggests that RL projections to PM[32] possibly exert inhibitory control via interneurons. PM and RD are strongly bi-directionally connected, but less so with anterior association areas[29,32], thus substantiating our functional findings. What could be reasons for the pronounced suppression of PM and RD? PM has been studied mostly in the context of visual tasks highlighting its role in spatial processing and navigation[48–50]. RD also connects to hippocampal regions, conveys top-down effects in a visual discrimination task[4], and is linked to spatial navigation and memory[51–53]. Therefore, under our experimental conditions where spatial navigation is not relevant, this network could be actively suppressed. Future studies may investigate whether in tasks distinct from ours, where spatial aspects are important, posterior association areas show enhanced activity whereas anterior areas including RL may be suppressed. Such suppression could be mediated through long-range inputs targeting inhibitory cell types[54–56] or long-range GABAergic projections originating from subcortical areas such as the hippocampus[57–59]. Alternatively, RL could be the association area for processing tactile information whereas PM is responsible for visual information and not needed in our task. We do not think that posterior and anterior network dissociation relates to retinotopic properties (posterior areas are more closely linked to the upper visual field[60]). First, we emphasize that we conducted experiments in the dark where the incoming texture was very hard to see. Second, specifically RL and PM, on average, refer to similar retinotopic positions, especially in terms of elevation[60]. Third, RL is more related to nearby visuo-tactile space, rather than a specific retinotopic position[61]. Fourth, we find learning-related modulations across individual days whereas the trial variables (i.e. position of incoming texture and illumination) were constant, making it less likely that retinotopic parameters primarily affected our results. In summary, our results highlight the distributed functional reorganization that cortical areas undergo during learning, progressing in two distinct major phases that reflect the initial transition into ready-for-learning modus and the subsequent establishment of the specific cortical flow pattern needed to solve the task.

## Methods
**Animals and surgical procedures**. Methods were carried out according to the guidelines of the Veterinary Office of Switzerland and following approval by the

Cantonal Veterinary Office in Zurich. A total of 7 adult male mice (1-4 months old) were used in this study. These mice were triple transgenic Rasgrf2-2A-dCre; CamK2a-tTA;TITL-GCaMP6f animals, expressing GCaMP6f in excitatory neocortical layer 2/3 neurons[11]. To generate triple transgenic animals, double transgenic mice carrying CamK2a-Tta[62] and TITL-GCaMP6f[63] were crossed with a Rasgrf2-2A-dCre line ([64]; individual lines are available from The Jackson Laboratory as JAX# 016198, JAX#024103, and JAX# 022864, respectively). The Rasgrf2-2A-dCre;CamK2a-tTA;TITL-GCaMP6f line contains a tet-off system, by which transgene expression can be suppressed upon doxycycline treatment[65,66]. However, doxycycline treatment is not necessary in these animals, since the Rasgrf2-2A-dCre line holds an inducible system of its own, given that the destabilized Cre (dCre) expressed under the control of the Rasgrf2-2A promoter needs to be stabilized by trimethoprim (TMP) to be fully functional. TMP (Sigma T7883) was reconstituted in Dimethyl sulfoxide (DMSO, Sigma 34869) at a saturation level of 100 mg/ml, freshly prepared for each experiment. For TMP induction, mice were given a single intraperitoneal injection (150 μg TMP/g body weight; 29 g needle; 3–5 days post-surgery), diluted in 0.9% saline solution.

We used an intact skull preparation[67] for chronic wide-field calcium imaging of neocortical activity[11]. Mice were anesthetized with 2% isoflurane (in pure $O_2$) and body temperature was maintained at 37 °C. We applied local analgesia (lidocaine 1%), exposed and cleaned the skull, and removed some muscles to access the entire dorsal surface of the left hemisphere (Fig. 2a; ~6 × 8 mm² from ~3 mm anterior to bregma to ~1 mm posterior to lambda; from the midline to at least 5 mm laterally). We built a wall around the hemisphere with adhesive material (iBond; UV-cured) and dental cement "worms" (Charisma). Then, we applied transparent dental cement homogenously over the imaging field (Tetric EvoFlow T1). Finally, a metal post for head fixation was glued on the back of the right hemisphere. This minimally invasive preparation enabled high-quality chronic imaging with high success rate.

**Texture discrimination task**. Mice were trained on a go/no-go discrimination task (Fig. 1a) using a data acquisition interface (USB-6008; National Instruments) and custom-written LabVIEW software (National Instruments[27]). Each trial started with an auditory cue (stimulus cue; 2 beeps at 2 kHz, 100-ms duration with 50-ms interval), signaling the approach of either two types of sandpapers (grit size P100: rough texture; P1200: smooth texture; 3M) to the mouse's whiskers as 'go' or 'no-go' textures (Fig. 1a; pseudo-randomly presented with no more than three repetitions). Sandpapers were mounted onto panels attached to a stepper motor (T-NM17A04; Zaber) mounted onto a motorized linear stage (T-LSM100A; Zaber) to move textures in and out of reach of whiskers. The texture stayed in touch with the whiskers for 2 s, and then it was moved out after which an additional auditory cue (response cue; 4 beeps at 4 kHz, 50-ms duration with 25-ms interval) signaled the start of a 2-s response period. The stimulus and response cues were identical in both textures. A water reward (~3 μL) was given to the mouse for licking for the go texture only after the response cue ('hit'), i.e. for the first correct lick during the response period (Fig. 1e; lick were detected using a piezo sensor). Punishment with white noise was given for licking for the no-go texture ('false alarms'; FA). Licking before the response cue was neither rewarded nor punished. Reward and punishment were omitted when mice withheld licking for the no-go ('correct-rejections', CR) or go ('Misses') textures. The licking detector remained in a fixed and reachable position throughout the entire trial. Note that the auditory tones merely served as cues defining the temporal trial structure, but had no predictive power with respect to the go or no-go condition. The first auditory tone signaled the trial-start and thus predicted the upcoming arrival of the texture as the task-relevant stimulus, whereas the second auditory tone indicated the availability of a water reward in the go trials. Licking before the response cue was allowed and did not lead to punishment or early reward.

**Training and performance**. Five mice were trained to lick for the P100 texture (mice #1-4 and 7) and 2 mice were trained to lick for the P1200 texture (mice #5 and 6). Mice were first handled and accustomed to head fixation before starting water scheduling. Before imaging began mice were conditioned to lick for reward after the go texture (presented within a similar trial structure as the task itself). Imaging began only after mice reliably licked for the response cue (typically after the first day; 200–400 trials). On the first day of imaging, mice were presented with the 'go' texture and after 50 trials the 'no-go' texture was gradually introduced (starting from 10% and increasing by 10% approximately every 50 trials[68]) until reaching 50% probability for the no-go texture by the end of the day. During the second day, most mice continuously licked for both textures (Supplementary Fig. 2). Thus after around 100 trials, we increased no-go probability to 80% and waited for mice to perform three continuous CR trials before returning to 50% probability. This was done for several times until mice increased their performance, specifically withheld licking for the no-go texture. In mice that still continued to lick for both textures we additionally repeated the wrong response until a correct response. In all mice, a 50% protocol was presented with no repetitions as soon as they reached expert level (d′ > 1.5). 6 out of the 7 mice learned the task within 3–4 days after around a thousand trials (Fig. 1d; Supplementary Fig. 2). Mouse #7 learned the task within 10 days. An effort was made to maintain a constant position

of the texture and cameras across imaging days in order to maintain similar stimulation and imaging parameters.

**Wide-field calcium imaging.** We used a wide-field approach to image large parts of the dorsal cortex while mice learned to perform the task[11]. A sensitive CMOS camera (Hamamatsu Orca Flash 4.0) was mounted on top of a dual objective setup. Two objectives (Navitar; top objective: D-5095, 50 mm f0.95; bottom objective inverted: D-2595, 25 mm f0.95) were interfaced with a dichroic (510 nm; AHF; Beamsplitter T510LPXRXT) filter cube (Thorlabs). This combination allowed a ~9-mm field-of-view, covering most of the dorsal cortex of the hemisphere contralateral to texture presentation. Blue LED light (Thorlabs; M470L3) was guided through an excitation filter (480/40 nm BrightLine HC), a diffuser, collimated, reflected from the dichroic mirror, and focused through the bottom objective ~100 μm below the blood vessels. Green light emitted from the preparation passed through both objectives and an emission filter (514/30 nm BrightLine HC) before reaching the camera. The total power of blue light on the preparation was <5 mW, i.e., <0.1 mW/mm². At this illumination power we did not observe any photobleaching. Data was collected with a temporal resolution of 20 Hz and a spatial sampling of 512 × 512 pixels, resulting in a spatial resolution of ~20 μm/pixel. On each imaging day a green reflectance image was taken as reference to enable registration across different imaging days using the blood vessel pattern (fiber-coupled LED illuminated from the side; Thorlabs).

**Mapping and area selection.** Each mouse underwent a mapping session under anesthesia (1% isoflurane), in which we presented five different sensory stimuli (contra-lateral side): a moving bar stimulating multiple whiskers, the forelimb paw, or the hindlimb paw (20 Hz for 2 s); visual stimulation with a blue LED in front of the eye (100 ms duration; approximately zero elevation and azimuth); and white noise auditory stimulation (2 s. duration). The averaged evoked maps clearly showed activation patches in the expected areas (Fig. 1c; Supplementary Fig. 1a). Next, we registered each imaging day to the mapping day using skull coordinates from the green images. Finally, we registered each mouse onto a 2D top view mouse atlas using both functional patches from the mapping and skull coordinates (Supplementary Fig. 1; ©2004 Allen Institute for Brain Science. Allen Mouse Brain Atlas. Available from: http://mouse.brain-map.org/[29]). Within the atlas borders, we defined 25 areas of interest, with some manual modifications within these borders to fit the functional activity for each mouse. Motor cortex areas were defined based on stereotaxic coordinates and functional patches for each mouse (see below). Thus all mice had similar regions of interest that were comparable within and across mice.

We grouped these 25 areas into auditory (green), association (pink), somatosensory + V1 (blue), and motor (red) areas (Fig. 1d and Supplementary Fig. 1b). Auditory areas: Primary auditory (A1), Auditory dorsal (AD) and Temporal association area (TEA). Sensory areas: Somatosensory mouth (Mo), Somatosensory nose (No), Somtosensory hindlimb (HL), Somtosensory forelimb (FL), Barrel cortex (BC; Primary somatosensory whisker); Secondary somatosensory whisker (S2), Somtosensory trunk (Tr) and Primary visual cortex (V1). Motor areas: whisker-related primary motor cortex (M1; 1.5 anterior and 1 mm lateral from bregma, corresponding to the whisker evoked activation patch in M1 from the mapping session), anterior lateral motor cortex (ALM; 2.5 anterior and 1.5 mm lateral from bregma[69]) and secondary motor cortex (M2; 1.5 anterior and 0.5 mm lateral from bregma corresponding[11]). Association cortex: Rostrolateral (RL), Anterior (A), Anterior lateral (AL), Anterior medial (AM), Posterior medial (PM), Lateral medial (LM), Lateral intermediate (LI), Posterior lateral (PL), Post-rhinal (PR), Retrosplenial dorsal (RD) and Retrosplenial angular (RA). We note that our definition of association cortex is broad and may include or exclude areas that are not necessarily classical association areas. In addition, we further divided association areas into anterior (RL, A, AM, and AL) and posterior (PM, L, LI, PL, PR, RD, and RA) association cortex (dashed red line in Fig. 6d).

**Control experiments.** In control experiments, we excluded confounding effects of autofluorescence or non-calcium-related intrinsic signals, by exciting the wide-field preparation with green light, showing no positive responses during cue-, pre-, and stim-period (Supplementary Fig. 11; For additional controls for non-calcium related optical signals see Gilad et al.[11]). Therefore, in the experiments presented in this study non-calcium-related intrinsic signals have no major influence on the GCaMP6f signals, especially in the cue- and pre-periods. To control for possible changes in responses across several days that are not necessarily related to learning, we evaluated the stability of areal activity in expert mice imaged across 5 consecutive days. Responses in BC (during stim-period), RL (during pre-period), and A1 (during cue-period) across 5 days were relatively flat (n = 4 mice). In addition, trial-shuffled data across learning eliminated these changes in responses and resulted in a relatively flat change in response (10³ iterations). Taken together, changes in activity across a learning period of several days is more likely to be learning related rather than day-to-day fluctuations in activity.

**Whisker and body tracking.** In addition to wide-field imaging, we tracked movements of the whiskers and the body of the mouse during the task (Fig. 1a). The mouse was illuminated with a 940-nm infrared LED. Whiskers were imaged at 50 Hz (500 × 500 pixels) using a high-speed CMOS camera (A504k; Basler), from

which we calculated time course of whisking envelope and the time of first touch (see below). An additional camera monitored the movements of the mouse at 30 Hz (The imaging source; DMK 22BUC03; 720 × 480 pixels). We used movements of both forelimbs and the head/neck region to assess body movements, to reliably detect large movements (Fig. 1a; see Data Analysis below). Importantly, mice performed this task in the dark where motor parameters were collected using infrared light. The only light in the setup was the blue illumination pattern that was focused through the second objective onto the wide-field preparation. Illumination conditions were still very low despite this light and mice could perform the task in complete darkness. Thus, it is unlikely that visual cues from the incoming textures could affect responses in different association areas.

**Data analysis.** Data analysis was performed using Matlab software (Mathworks). All mice were continuously imaged during learning (5–11 days). Wide-field fluorescence images were sampled down to 256 × 256 pixels and pixels outside the imaging area were discarded. This resulted in a spatial resolution of ~40 μm/pixel and was sufficient to determine cortical borders, despite further scattering of emitted light through the tissue and skull. Each pixel and each trial were normalized to baseline several frames before the stimulus cue (frame 0 division). In this study, we grouped trials based on the texture type, i.e. go or no-go texture (see Calculating learning curves below). We defined three time periods within the trial structure: cue (−1.9 to −1.6 relative to texture stop), pre (−1 to 0.5 s relative to texture stop), and stim (−0.5 to 0.5 relative to texture stop; Fig. 1d). Naïve and expert phases were defined as the first and last 500 trials, respectively. Responses during the first 50 trials were similar to the first 500 trials (p > 0.05; Signed rank test across mice), indicating that the very early stages of stimulus presentation were not substantially different.

**Calculating body movements.** We used a body camera to detect general movements of the mouse (30 Hz frame rate; Supplementary Fig. 1a). For each imaging day, we first outlined the forelimbs and the neck areas (one area of interest for each), which were reliable areas to detect general movements. Next, we calculated the body movement (1 minus frame-to-frame correlation) within these areas as a function of time for each trial. Thresholding at 3 s.d. (across time frames before stimulus cue) above baseline (defined as the 5th percentile) resulted in a binary movement vector (either 'moving' or 'quiet') for each trial[11].

**Whisker tracking and first-touch analysis.** The average whisker angle across all imaged whiskers was measured using automated whisker tracking software[70]. The mean whisker envelope was calculated as the difference between maximum and minimum whisker angles along a sliding window equal to the imaging frame duration (50 ms; refs. [11,27]). Whisker envelope was normalized just before the auditory cue similar to wide-field data (Frame zero). In addition, we manually detected the first frame, in which any whisker touched the upcoming texture, using the movies from the whisker camera (LabVIEW custom program). The first touch occurred on average 0.33 and 0.34 s before the texture stopped for naïve and expert mice respectively. Time of first touch did not differ between expert and naïve mice (P > 0.05; Mann–Whitney U-test; n = 7 mice). We note that the first touch occurred mostly (but not exclusively) in the pre-period from −1 to −0.5 relative to texture stop.

**Calculation of curves across learning.** Trials were binned (n = 50 trials with no overlap) across learning and the performance (defined as d′ = Z(Hit/(Hit +Miss)) − Z(FA/(FA + CR)) where Z denotes the inverse of the cumulative distribution function) was calculated for each bin. Next, each behavioral learning curve was fitted with a sigmoid function

$$S(t) = a \frac{1}{1 + e^{\frac{-(t-b)}{c}}} \quad (1)$$

Where a denotes the amplitude, b the time point (in trial numbers) of the inflection point, and c the steepness of the sigmoid. A d′ = 1.5 was defined as the threshold and mice were ordered based on the trial number at which they crossed threshold (i.e. learning threshold; Fig. 1g). Varying the d′ threshold maintained the order of the mice based on their learning threshold (see Fig. 1f).

To compare the behavioral learning curve with other behavioral parameters and neuronal activity, we similarly grouped trials and separated them based on the texture type, i.e. hit and miss trials were grouped into the go texture trials; CR and FA trials were grouped into the no-go texture trials. Our main focus in this study was on the go texture (presented in Figs. 2–6). Therefore, stimulus identity was kept similar across learning. However, results were maintained when considering only the no-go texture trials. Only in Fig. 7 we compare between go and no-go textures to calculate discrimination power. Next, we can present the dynamics of a behavioral parameter (i.e. body movement, whisking envelope or licking probability) or cortical area activity (averaged over pixels) in two-dimensional temporal spaces where the x-axis is the trial temporal structure (i.e. trial dimension) and the y-axis is the learning profile across trials and days (i.e. learning dimension; for examples see Figs. 2a, e, i and 3c (top)). From this 2D temporal space we could average across trials of the learning dimension, e.g. during naïve and expert states (for example see Figs. 2b, f, j and 3c (middle)). Alternatively, we

can average across time frames within the trial dimension, to obtain a response curve across learning for a specific time period (i.e. cue-, pre-, or stim-period; additionally smoothed with a Gaussian kernel ($2\sigma = 9$) and fitted with a sigmoid function; for example see Figs. 2c, g, k and 3c (bottom)). Thus we are able to obtain a curve across learning for a specific area or behavioral parameter which are comparable to the behavioral learning curve of the mouse. The sigmoid fits of the response curves from different cortical areas were normalized between 0 and 1 in order to compare between response curves of different areas. This was done mainly because of the different activation ranges across learning for each area. Non-normalized learning curves are presented in Supplementary Fig. 7. In an additional analysis we also fitted each response curve for all areas and time periods with a double sigmoid fit in order to fit both the initial suppression and the later enhancement that was present in some curves (e.g. Fig. 4d; Supplementary Fig. 9):

$$d(t) = a1\left(1 - \frac{1}{1 + e^{\left(\frac{-(t-b1)}{c1}\right)}}\right) + a2\left(1 + \frac{1}{1 + e^{\left(\frac{-(t-b2)}{c2}\right)}}\right) + d \qquad (2)$$

with $a1$ and $a2$ as amplitudes, $b1$ and $b2$ as inflection points (in trial numbers), and $c1$ and $c2$ as steepness parameters of the descending and ascending sigmoid, respectively. $d$ is a baseline parameter, which was set to the minimum value of a curve. Thus, for each area we could quantify the amount (amplitude) and timing (latency) of both suppression and enhancement during each time period relative to the learning threshold. Finally, to quantify the enhancement-suppression ratio we calculated the modulation index (MI) as

$$MI = \frac{a2 - a1}{a2 + a1} \qquad (3)$$

ranging from $-1$ and 1, with positive values indicating more enhancement, negative values indicating more suppression, and near zero values indicating similar amounts of suppression and enhancement.

**Calculating learning maps and seep maps**. To study the relationship between the behavioral learning curve and the learning curves of all pixels we calculated a 'learning map' (Fig. 5). This was done by calculating the correlation coefficient (r) between the behavioral learning curve of the mouse and the learning-related $\Delta F/F$ changes of each pixel (Fig. 5a). This can be done for a specific time period (i.e. cue-, pre- or stim-period; Fig. 5a, b) or for each time frame (Fig. 5c). To calculate the relationship between the learning-related $\Delta F/F$ changes of a specific area (i.e. seed) and the learning-related $\Delta F/F$ changes of all pixels we calculated a seed correlation map (Fig. 6). This was done similarly to the learning map by only substituting the behavioral learning curve with the learning-related $\Delta F/F$ changes of the desired area (defined as the seed area; Fig. 6a). We chose seed areas to be RL, PM, and RD which were of the highest interest based on previous analysis and best represent the main trends of neuronal changes during learning. A full correlation matrix between all learning curves is presented in Supplementary Fig. 10.

**Discrimination power between go and no-go texture**. To measure how well could neuronal populations discriminate between go and no-go textures, we calculated a receiver operating characteristics (ROC) curve and calculated its area under the curve (AUC; with a value of 0.5 indicating no discrimination power). This can be done for each pixel (Fig. 7e), each area (Fig. 7d), each time frame (Fig. 7b), and across learning (Fig. 7f). We put our main focus on the stim-period when the texture touched the whiskers. To calculate significance, we calculated the sample distribution by trial shuffling between go and no-go textures ($n = 100$ iterations). Exceeding mean ± 2 s.d. of the sample distribution is defined as significant (Fig. 7b).

**Statistical analysis**. In general, non-parametric two-tailed statistical tests were used, Mann–Whitney $U$-test to compare between two medians from two populations or the Wilcoxon signed-rank test to compare a population's median to zero (or between two paired populations). Multiple group correction was used when comparing between more than two groups.

**Reporting summary**. Further information on research design is available in the Nature Research Reporting Summary linked to this article.

## Data availability
The data and code that support the findings of this study are available from the corresponding author upon reasonable request.

## Code availability
The codes used to analyze the data of current study are available from the corresponding authors on reasonable request.

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

## Acknowledgements

This work was supported by grants from the Swiss National Science Foundation (SNSF) (31003A-149858 and 310030B_170269; F.H.), the European Research Council (ERC Advanced Grant BRAINCOMPATH, project 670757; F.H.), an Edmond and Lily Safra Center for Brain Sciences (ELSC) postdoctoral fellowship (A.G.), an EMBO long-term postdoctoral fellowship (ALTF_1077-2014; A.G.). This project has received funding from the European Union's Horizon 2020 research and innovation program under the Marie Skłodowska-Curie grant agreement No 659719 (A.G.).

## Author contributions

A.G. and F.H. designed the experiments. A.G. conducted the experiments. A.G. and F.H. performed data analysis. A.G. and F.H. wrote the manuscript.

## Competing interests

The authors declare no competing interests.
