## [Peer Review File · Nature Communications]

Reviewers' Comments:

Reviewer #1:

Remarks to the Author:

Manuscript # NCOMMS-19-26125-T

Spatiotemporal refinement of signal flow through association cortex during learning
by Ariel Gilad and Fritjof Helmchen

In their study Gilad and Helmchen have investigated changes in the response characteristics of different cortical areas (associative and sensory) during task learning using wide-field calcium imaging in an intact mouse skull preparation expressing GCaMP6f in layer 2/3 pyramidal neurons of the neocortex. The authors examine changes in cortical activity patterns during the learning process. They found spatiotemporal changes in signal processing, namely that the activity of layer 2/3 pyramidal neurons in the association cortex is suppressed at the pre-learning stage; with ongoing learning this initial suppression changes and an enhancement of the activity A1 cortex and subsequently the rostra-lateral part of the association cortex (which was suppressed at the pre-learning stage) and finally the S1 barrel cortex. The findings are novel and expand our knowledge on brain-wide changes in the activity of different cortical areas during task learning.

While the manuscript is certainly of high quality it suffers from a number of imprecision such as errors in the figures and figure legend as well as inconsistencies or plain errors in the reference list). The authors should carefully proofread their manuscript before resubmission.

Specific points:

Methods

How high was the spatial resolution of the imaging set-up. Did this allow for exact determination of cortical areas and what was the margin of error? Furthermore, how large was the area for wide-field imaging?

Do all layer 2/3 pyramidal neurons (i.e. in every cortical area under study) express GCaMP6f or is anything known about a selective expression pattern?

Results

Page 7

The authors write define 'expert' phase more clearly, in particular with respect to the time point measured (last 500 sweeps of what?).

Page 7-8 and Figure 1f and Supplementary figure 2a, b

There is quite some variability in the pre-learning and learning curves; for some mice it is very steep while it increases only very gradually in others.

Page 11 and Figure 3a-c

Even for the two example mice shown in this figure there appears to be a large degree of variability in the 'cue', 'pre' and 'stim' response, both in the naïve and expert phase. For example, the response in the naïve m3 mouse is markedly smaller than that in the m6 mouse; also the expert m6 mouse shows a strong signal in some motor cortical region. Could the authors please comment on this variability. From Figure 3 and also Supplementary figure 7 a strong enhancement of motor cortex activity is shown for the 'stim' phase but this is not commented on in the Discussion.

Figure 5c

The m6 mouse shows a very late enhancement in A1 cortex during pre-learning/learning. Could the authors please comment?!

Page 20, line 408

In the main text the authors refer the reader to Figure 6f. Surely, this should be 7f.

Discussion

Figure 8

The labeling in Figure 8; appears to be wrong. I assume Phase 2 and Phase 1 should be switched.

Page 24, 1st paragraph

Could the authors please be more specific regarding the 'pronounced reorganization during this training phase'? What is actually changing? Do they mean the excitability, the synaptic connectivity or the axonal projection pattern of L2/3 pyramidal neurons?

Page 24, 2nd paragraph

Does the paper really show a unidirectional projection from RL to PM? The authors of the quoted paper (Wang et al., 2012) state that 'a network with two modules, indicating that medial/anterior extrastriate visual areas (AL, RL, A, AM, PM) are more strongly linked to parietal, motor, and limbic cortices, whereas lateral extrastriate areas (LM, P, LI, POR) are preferentially connected to temporal and parahippocampal regions.' Furthermore, the authors of that paper also wrote that 'Unlike previous studies, which showed that all corticocortical connections in rat are reciprocal (...), we found 20 – 40% unidirectional connections. We believe that this is a gross overestimate because cell body labeling with BDA was extremely sparse and iontophoretic injections clearly suboptimal for retrograde tracing of reciprocal connections (...).' This calls the assumption of the authors of this study in question. Please comment!

On another note, could the authors please comment/speculate whether a 'reorganization' of signaling between different cortical areas would be similar for deeper cortical layers. Is a different reorganization possible? How would subcortical structures affect learning in cortical layer 2/3. While I do not suggest to perform additional experiments here I think it would be relevant to discuss these points.

References list

There are several errors in the reference list (see below). In addition, the referencing style is not uniform, e.g. the page numbers are sometime given in full (i.e. 1001-1010) and sometimes in short version (i.e. 1001-10). Please correct to Journal style.

References 9: No volume number

Reference 11: Volume is given as 'Neuron 0'; page numbers are missing

Reference 15: No volume number

Reference 26: Journal name is missing

Reference 33: Journal is given as 'Science (80-.)' but should read just 'Science'

Reference 43: Journal is given as 'Science (80-.)' but should read just 'Science'

Reference 51: Journal name is missing; however, the authors give the dot here but not for all other references. Please correct to journal style

Supplementary figures

Supplementary figure 1 legend

The legend refers to three panels (a-c); the figure shows only two (a, b). It appears that panel b and c in the legend refer to panels a and b of the figures, respectively. Also, in panel b (c in the legend) auditory cortex is shown separately while visual and somatosensory cortical areas are summarized as 'sensory' cortex. This is somewhat confusing; please give a rationale here!

Reviewer #2:

Remarks to the Author:

NCOMMS-19-26125-T

Spatiotemporal refinement of signal flow through association cortex during learning.

Gilad & Helmchen present an interesting and detailed description of cortical activation during a texture discrimination task, showing activation of a complex and stereotyped sequence of cortical areas and the development of this sequence during learning. Activation is distributed, as in several recent studies of go/no-go behaviors, but the task is different and the sequence of activation is therefore distinct. Gilad & Helmchen employ only widefield imaging and can provide no information on which cell populations are responsible for the activity or on whether these cortical areas play causal roles in driving discrimination behavior. And yet this focus on widefield imaging is an asset since permits extensive analyses that are carefully described by the authors, bringing remarkable clarity to an extremely complex topic. This paper would be an excellent guide for future studies and I expect it will prove influential.

Major

(1) While I consider the imaging-only approach an asset, the authors could perhaps dedicate more of the discussion to considering whether the observed patches of activity play causal roles in driving the behavior. There's much information in the literature about the roles of these regions. Perhaps the authors can combine their imaging results and the literature to make predictions about causal roles that would be testable in future perturbation experiments.

(2) The authors describe trial-averaged activity, almost exclusively. I imagine there's substantial trial-to-trial variability and expect their imaging experiments provide sufficient signal-to-noise to measure this variability. Some analysis of trial-to-trial variability might provide further insights.

Minor

(3) Results, lines 95-96: 'they were neither rewarded nor punished when they withheld licking for the go and no-go textures ('correct-rejections', CR, and 'Misses', respectively).' I believe CR and Misses are transposed: a correct rejection must be withholding licking for a no-go texture, and a miss withholding for a go texture.

(4) Discussion, starting line 487: The authors have perhaps overlooked the retinotopic biases of higher visual areas in their discussion of signaling between anterior and posterior visual areas. Might whisker contacts with the textured surface occur below the horizontal meridian, thereby evoking activity in anterior higher visual areas, but not posterior higher visual areas?

(5) Methods, line 536: At what age was the intraperitoneal injection administered?

(6) Methods, line ~557: The authors should provide some information on the lick detector (image-based? IR beam break? capacitive? piezo?). Do they have estimates of the failed detection and false detection rates?

(7) Supplementary figure 1 legend needs revision. The description of a schematic in panel a needs to

be deleted. Panel b describes maps 'for two example mice' when maps are provided for all 7 mice.

Reviewer #3:

Remarks to the Author:

Gilad and Helmchen present a well-written and comprehensive analysis of cortex-wide changes in brain activity across the learning trajectory. Presentation of data from multiple animals with diverse learning rates and variable responses is important, as there is considerable heterogeneity of responses across different areas (for example, some animals show a strong cue response and others do not). Capturing this biological variability is an important addition to the field. The authors show that suppression of activity in RL and PM, two areas related to visual processing, can predict learning curves hundreds of trials in advance. This is potentially the most important and interesting finding, but its significance is not adequately discussed. Otherwise, the experimental design and analysis are clear and convincing and carried out to a very high standard. This paper should thus be an important addition to the field.

1. It appears that BC activity is enhanced prior to the stimulus phase, specifically in expert animals. What is the cue for this? This result is not adequately emphasized or discussed. Also, how many trials are averaged for the naïve animals? It still seems like there is some pre-stim activity in BC before the stim arrives – this may be an early indication of expectation or plasticity in the circuit. When exactly is the BC activity initiated? How does this timing change with learning? Can the naïve period be restricted to fewer trials, at the very earliest stages of presentation? Are the mice preconditioned in some way?

2. The major finding presented is that some (apparently) visual-associated areas (PM and RL) are suppressed as the animals develop expertise in the task. Because the tactile stimulus has visual characteristics (it swings into place, and the animals appear to be in a lighted area so that whisker touch data can be collected), it may be that the mouse is learning to disregard the visual aspects of the stimulus (since the texture plate swings into place in both types of trials) and instead focus on the tactile aspect of the stimulus. Thus, the downregulation of activity in these areas may not be a canonical feature of sensory learning (or of tactile learning more specifically), but rather may be related to this particular behavioral paradigm which includes both visual and tactile stimuli. This should be explicitly discussed and evaluated. Because the experiments and the analysis have been rigorously implemented, and the data presented are very clear, I do not think that this necessarily requires new experiments. However, this caveat should be discussed in both the abstract and the body of the paper.

3. Please speculate what the neurophysiological mechanisms are for enhancement of the go (stimulus-rewarded) response versus the no-go response in S1. This must be due to input from some other brain area? Surely you are not picking up differences in direct sensory input that correspond to specialized neural circuits for texture discrimination?

How can BF activity be increased for the Go texture prior to the texture presentation? Are the mice getting some auditory cue about what the texture will be, as the rotor moves into place?

4. By pairwise correlation, do decreased RD and increased RL activity provide better predictors of the inflection point than BC and RM changes (this is what appears to be the case in Figure 4a)? If you just use those two features, how good is the inflection point prediction? Why does Figure 4B plot "steepest point" and not some minimum $\Delta F/F$, since that seems to be more directly related to transitions in both performance and also activity in BC and RL? It's almost as if RD has a net inhibitory influence over RL – what is known about this circuitry?

5. The discussion about why PM and RL are suppressed is not adequate. Especially because this is a main finding, the authors are invited to speculate more about the significance of the observation, how it is related to prior studies in NHPs, and neural structures that might be critical to generate this suppression (they mention that there is inhibition – but, is that typical? That there is inhibitory input

from one area to the next?).

Minor

1. There is a considerable amount of jargon in the manuscript that impedes readability. Cf "inflection point" "learning map" "Pre-period" In Figure 3c, the red line (in the $\Delta F/F$ histograms) indicates the first whisker touch in a given trial. What is the dashed cyan line for? It may be misleading to suggest it is the stim, as the stim must be present for first touch to occur. If it is the center of the stimulus presentation time, it is confusing for the reader. In any case, it should be defined in each legend where it is used.
2. Figure 4d, e might be better as a Supplemental Figure, as the changes in A1 activity are interesting but rather unexplained.
3. In Figure 5, it appears that many (most?) brain areas are correlated with learning in the stim period. The special focus on BF is hard to interpret given this widespread effect.
4. Figure 5a is very complicated, and this analysis seems critically dependent upon registration of a given pixel between imaging sessions. What is the spatial extent of a given pixel and how closely can they be aligned across days?
5. For Figure 6f, what is the justification for separating anterior and posterior association areas?
6. On p13, please define "inflection point" for the normalized curves here. This is a key point and the significance of the observations are lost in the jargon
7. Please speculate what the neurophysiological mechanisms are for enhancement of the go (stimulus-rewarded) response versus the no-go response in S1. This must be due to input from some other brain area? Surely you are not picking up differences in direct sensory input that correspond to specialized neural circuits for texture discrimination?
How can BF activity be increased for the Go texture prior to the texture presentation? Are the mice getting some auditory cue about what the texture will be, as the rotor moves into place?
8. In Figure 8a, why is "phase 2" (adjacent to "naïve") above "phase 1" on the y-axis of the schematic?
9. What is "obsessive licking" (line 478-9)?
10. Line 96, add 'Fig. 1e' to help readers find the task design.
11. Figure 1e: Can you describe exactly when water was delivered and for how long?
12. Line 219, Supplementary Fig. 5: Is this the right figure? Supplementary Fig. 5 is across many trials over days, whereas the sentence seems to be talking about sequential activity within a trial.
13. Fig. 3b: Can we see the same graph averaged over all animals? Also including RD and PM?
14. Fig 3b: Please add descriptions about the black histogram inset in the legends.
15. Fig. 3d, pre: Why LI is highlighted in light purple instead of PM? (This appears to be a mistake)
16. Methods: Why do you use triple transgenic mice if you don't actually need or use the tet-off system?
17. Fig 8: Are phase 1 and phase 2 switched in the y-axis?
18. Fig 8b: Is this about physical movement response of mice? I think instead of this, a schematic graph similar to Fig 3b including the key regions (A1, RD, PM, RL, BC) would be a good visualization of the key findings.
19. Methods: Please add more description about TMP injections. How many injections per mice and exactly when?

Response to reviewers:

Manuscript # NCOMMS-19-26125-T

Spatiotemporal refinement of signal flow through association cortex during learning

by Ariel Gilad and Fritjof Helmchen

Reviewer #1 (Remarks to the Author):

In their study Gilad and Helmchen have investigated changes in the response characteristics of different cortical areas (associative and sensory) during task learning using wide-field calcium imaging in an intact mouse skull preparation expressing GCaMP6f in layer 2/3 pyramidal neurons of the neocortex. The authors examine changes in cortical activity patterns during the learning process. They found spatiotemporal changes in signal processing, namely that the activity of layer 2/3 pyramidal neurons in the association cortex is suppressed at the pre-learning stage; with ongoing learning this initial suppression changes and an enhancement of the activity A1 cortex and subsequently the rostra-lateral part of the association cortex (which was suppressed at the pre-learning stage) and finally the S1 barrel cortex. The findings are novel and expand our knowledge on brain-wide changes in the activity of different cortical areas during task learning.

While the manuscript is certainly of high quality it suffers from a number of imprecision such as errors in the figures and figure legend as well as inconsistencies or plain errors in the reference list). The authors should carefully proofread their manuscript before resubmission.

Thank you for your comments. Please see below our point-to-point response:

Specific points:

Methods

1) How high was the spatial resolution of the imaging set-up. Did this allow for exact determination of cortical areas and what was the margin of error? Furthermore, how large was the are for wide-field imaging?

In the data set collected the field of view was $\sim 9 \times 9 \text{ mm}^2$ and our pixel resolution was 512×512 resulting in a spatial resolution of approximately $\sim 20 \text{ }\mu\text{m}/\text{pixel}$. During data analysis we downsampled the data to 256×256 pixels, resulting in a spatial resolution of $\sim 40 \text{ }\mu\text{m}/\text{pixel}$. The minimal size of a cortical area is $300 \text{ }\mu\text{m}$ (lateral intermediate area, LI, along the medial-lateral axis, i.e., 7-8 pixels wide). One has to realize, however, that the effective spatial resolution is reduced because both excitation light and calcium indicator fluorescence signals are scattered through the tissue and the skull, which blurs signals and degrades the spatial resolution. We estimate that signals can be blurred across even a few hundred micrometers. However, since our method for registration to the Allen atlas relied on multiple anatomical and signal-based landmarks spread across several millimeters, we are confident that our area determination is precise enough (to within a few hundred micrometers) to extract the areal activity signals. We have now clarified these issues in the main text and the methods.

The revised text now reads (pg. 34): “Data was collected with a temporal resolution of 20 Hz and a spatial sampling of 512x512 pixels, resulting in a spatial resolution of approximately 20 $\mu\text{m}/\text{pixel}$ ”

And also in (pg. 36): “Wide-field fluorescence images were sampled down to 256x256 pixels and pixels outside the imaging area were discarded. This resulted in a spatial resolution of $\sim 40 \mu\text{m}/\text{pixel}$ and was sufficient to determine cortical borders, despite further scattering of emitted light through the tissue and skull.”

2) Do all layer 2/3 pyramidal neurons (i.e. in every cortical area under study) express GCaMP6f or is anything known about a selective expression pattern?

To the best of our knowledge GCaMP6f is expressed in the vast majority of excitatory L2/3 neurons in this line. This can be seen in the relevant transgenic characterization data from the Allen Institute (<http://connectivity.brain-map.org/transgenic/experiment/313182284>), showing strong expression of GCaMP6f in the large majority of layer 2/3 neurons homogenously throughout the neocortex. Given the CaMKII driver these cells should represent nearly all excitatory L2/3 neurons whereas inhibitory neurons and astrocytes should not be labeled. This expression pattern with a very high fraction of L2/3 neurons expressing indicator was also observed in the study Madisen et al. (ref. 56), to which we contributed. In addition to strong labeling of L2/3 neurons, scattered neurons in deeper cortical layers also show some expression, as well as some neuronal populations in deeper brain regions such as the hippocampus and hypothalamus. We do not presume that these very deep neurons contribute to our wide-field calcium signals.

Results

3) Page 7

The authors write define 'expert' phase more clearly, in particular with respect to the time point measured (last 500 sweeps of what?).

The expert phase is defined as the last 500 trials, during which a mouse performed the task with high performance ($d' > 1.5$). This definition was done for each mouse separately. This expert phase corresponds to the experimental phase when animal performance typically has reached a high plateau level (see learning curves in Fig. 1). Throughout the manuscript, when we refer to the ‘expert’ phase, we consider activity averaged over these last 500 trials (regardless of trial outcome).

The revised text now reads (pg. 7): “Naïve’ and ‘expert’ phases were defined as the first 500 and the last 500 trials imaged for each mouse, respectively.”

4) Page 7-8 and Figure 1f and Supplementary figure 2a, b

There is quite some variability in the pre-learning and learning curves; for some mice it is very steep while it increases only very gradually in others.

This is true. We believe this reflects the natural variability across mice in engaging in this type of task.

The revised text now reads (pg. 8): *“In addition, some mice displayed a steep learning curve whereas others showed more gradual learning, probably reflecting the natural variability of task engagement across mouse individuals.”*

5) Page 11 and Figure 3a-c

Even for the two example mice shown in this figure there appears to be a large degree of variability in the ‘cue’, ‘pre’ and ‘stim’ response, both in the naïve and expert phase. For example, the response in the naïve m3 mouse is markedly smaller than that in the m6 mouse; also the expert m6 mouse shows a strong signal in some motor cortical region. Could the authors please comment on this variability. From Figure 3 and also Supplementary figure 7 a strong enhancement of motor cortex activity is shown for the ‘stim’ phase but this is not commented on in the Discussion.

The reviewer is correct. There is a large variability of responses across mice for different cortical areas and different time periods. Some of this variability can be at least in part explained by the variable movement parameters for each mouse (Figs. 2 and Supplementary Fig. 3). For example, m6 mouse whisks more than m3 in the expert phase during the stim period (Supplementary Fig. 3b). Therefore, it is reasonable that the activation map during the stim period for m6 displays high activity in frontal whisker motor cortex (Fig. 3a). For these reasons, we also did not put a major emphasis on the stim period but rather focused more on the cue and pre periods, where the mice did not move much. In contrast, other differences between mice are harder to relate to simple motor differences, for example differences in responses in A1 during the cue period.

The revised text now reads (pg. 14): *“From these two example mice it is also evident that wide-field cortical activity varies between mice. Some of this variability can be explained by differences in movement parameters. For example, mouse #6 in the expert phase whisked more during the stim period compared to mouse #3 (Supplementary Fig. 3b), which may result in enhanced activity in frontal whisker motor cortex for mouse #6 (Fig. 3a). Other differences, such as for example variable A1 responses during the cue period, cannot be explained by simple motor parameters and may reflect intrinsic differences between mice.”*

6) Figure 5c

The m6 mouse shows a very late enhancement in A1 cortex during pre-learning/learning. Could the authors please comment?!

Figure 5c displays the correlation of the learning curve with the curve of learning-related $\Delta F/F$ changes for several areas, among them A1 (green line). For m6, A1 display a late enhancement (around 0.5 sec after texture stop, after the stim period), which means the activity in A1 during this time is positively correlated with the learning of the mouse. We have no convincing explanation for this, except for highlighting again the inter-mouse variability both in activity maps and motor parameters. It could be that m6 during this time moved less during the expert phase, which may lead to enhanced activity in A1 (Schneider, ..., Mooney 2014). As mentioned above, the increasing difficulties in interpreting calcium signals in the late sensation periods when animal enhance their movements and prepare for licking led us to refrain from a deeper analysis of these late time periods.

7) Page 20, line 408

In the main text the authors refer the reader to Figure 6f. Surely, this should be 7f.

Fixed.

Discussion

8) Figure 8

The labeling in Figure 8; appears to be wrong. I assume Phase 2 and Phase 1 should be switched.

Fixed.

9) Page 24, 1st paragraph

Could the authors please be more specific regarding the ‘pronounced reorganization during this training phase’? What is actually changing? Do they mean the excitability, the synaptic connectivity or the axonal projection pattern of L2/3 pyramidal neurons?

The revised text now reads (pg. 27): *“This result points to a pronounced reorganization during this training phase that may involve several factors such as inhibitory effects, excitation-inhibition balance, synaptic plasticity or top-down interactions.”*

10) Page 24, 2nd paragraph

Does the paper really show a unidirectional projection from RL to PM? The authors of the quoted paper (Wang et al., 2012) state that ‘a network with two modules, indicating that medial/anterior extrastriate visual areas (AL, RL, A, AM, PM) are more strongly linked to parietal, motor, and limbic cortices, whereas lateral extrastriate areas (LM, P, LI, POR) are preferentially connected to temporal and parahippocampal regions.’ Furthermore, the authors of that paper also wrote that ‘Unlike previous studies, which showed that all corticocortical connections in rat are reciprocal (...), we found 20 – 40% unidirectional connections. We believe that this is a gross overestimate because cell body labeling with BDA was extremely sparse and iontophoretic injections clearly suboptimal for retrograde tracing of reciprocal connections (...).’ This calls the assumption of the authors of this study in question. Please comment!

We have omitted the claim for uni-directional projection from RL to PM (pg. 28).

11) On another note, could the authors please comment/speculate whether a ‘reorganization’ of signaling between different cortical areas would be similar for deeper cortical layers. Is a different reorganization possible? How would subcortical structures affect learning in cortical layer 2/3. While I do not suggest to perform additional experiments here I think it would be relevant to discuss these points.

We have now added an additional discussion relating to this relevant point. The revised text now reads (pg. 27): *“Continuing this line, in this study we focused on cortex-wide layer 2/3 excitatory neurons, but learning-related dynamics may involve other circuit elements such as deep cortical*

layers^{37,38}, inhibitory subtypes³⁹ or subcortical areas^{38,40}. For example, activity of layer 6 neurons was shown to decrease the gain modulation of superficial layers³⁷, which may lead to the initial global suppression in association areas. Another possibility is that higher-order thalamocortical connections may drive synaptic plasticity during learning³⁸ and may be specific to distinct association areas such as RL⁴⁰. Specifically, these factors may also contribute to the enhanced discrimination between go and no-go trials in BC for expert mice, in which a possible integration of anticipatory signals from RL and higher-order thalamocortical inputs may enhance the population activity in BC for ‘go’ textures. Future studies should expand our observations to other cortical layers, cell-types, and brain areas in order to gain a more comprehensive understanding of learning-related mechanisms.”

12) References list

There are several errors in the reference list (see below). In addition, the referencing style is not uniform, e.g. the page numbers are sometime given in full (i.e. 1001-1010) and sometimes in short version (i.e. 1001-10). Please correct to Journal style.

References 9: No volume number

Reference 11: Volume is given as ‘Neuron 0’; page numbers are missing

Reference 15: No volume number

Reference 26: Journal name is missing

Reference 33: Journal is given as ‘Science (80-.)’ but should read just ‘Science’

Reference 43: Journal is given as ‘Science (80-.)’ but should read just ‘Science’

Reference 51: Journal name is missing; however, the authors give the dot here but not for all other references. Please correct to journal style

Done.

Supplementary figures

13) Supplementary figure 1 legend

The legend refers to three panels (a-c); the figure shows only two (a, b). It appears that panel b and c in the legend refer to panels a and b of the figures, respectively. Also, in panel b (c in the legend) auditory cortex is shown separately while visual and somatosensory cortical areas are summarized as ‘sensory’ cortex. This is somewhat confusing; please give a rationale here!

The panel numbering has been corrected. In this study we compromised on separating auditory cortices (green) from other sensory cortices (blue). The blue areas were initially defined as ‘Somatosensory + primary visual’. That is V1 was grouped with somatosensory for simplicity. We now revert back to this naming of the blue areas as ‘Somatosensory + V1’ throughout the manuscript.

Reviewer #2 (Remarks to the Author):

NCOMMS-19-26125-T

Spatiotemporal refinement of signal flow through association cortex during learning.

Gilad & Helmchen present an interesting and detailed description of cortical activation during a texture discrimination task, showing activation of a complex and stereotyped sequence of cortical areas and the development of this sequence during learning. Activation is distributed, as in several recent studies of go/no-go behaviors, but the task is different and the sequence of activation is therefore distinct. Gilad & Helmchen employ only widefield imaging and can provide no information on which cell populations are responsible for the activity or on whether these cortical areas play causal roles in driving discrimination behavior. And yet this focus on widefield imaging is an asset since permits extensive analyses that are carefully described by the authors, bringing remarkable clarity to an extremely complex topic. This paper would be an excellent guide for future studies and I expect it will prove influential.

Thank you for your comments. Please see below our point-to-point response:

Major

(1) While I consider the imaging-only approach an asset, the authors could perhaps dedicate more of the discussion to considering whether the observed patches of activity play causal roles in driving the behavior. There's much information in the literature about the roles of these regions. Perhaps the authors can combine their imaging results and the literature to make predictions about causal roles that would be testable in future perturbation experiments.

We agree with the reviewer that these are relevant issues to discuss. We have now added a new paragraph in the discussion, aiming to connect between our observational data and the possible causal roles of the relevant cortical areas based on previous literature (pg. 27): *“This study focuses on wide-field observations across the time course of learning, but the causal role of each area during specific time periods remains unknown. Silencing BC during sensation in expert mice has been shown to impair discrimination and detection performance (refs. ⁴¹⁻⁴⁴; but see ref. ⁴⁵), highlighting the role of BC in texture discrimination. But are the observed pre-stimulus cortical dynamics necessary for discrimination learning? RL was found to be causally linked to performance during different discrimination tasks during the stimulus period^{16,18,46,47}, but not much is known about the role of RL during pre-stimulus periods. RL (as part of posterior parietal cortex) was also causally linked to maintaining history-dependent information from previous trials, highlighting its possible involvement in pre-stimulus periods²³. As for the initial suppression in PM and RD, we speculate that activating these areas during pre-learning (resulting in a uniform activation across association areas), may counteract sharpening of activity in association areas and thereby prolong the time to learning onset. In contrast, creating a salient activation pattern across association areas (e.g. enhancing RL while suppressing PM and RD) in naïve mice, may facilitate learning. Future studies will focus on the effects of different association areas during pre-learning periods.”*

(2) The authors describe trial-averaged activity, almost exclusively. I imagine there's substantial trial-to-trial variability and expect their imaging experiments provide sufficient signal-to-noise to measure this variability. Some analysis of trial-to-trial variability might provide further insights.

We definitely agree that single-trial analysis may provide further insights. We would like to point out the single-trial ROC analysis we performed to discriminate between single Hit and CR trials (Figure 7; we added some additional text emphasizing that this analysis is at the single-trial level). We have now added additional single-trial analysis, directly comparing the trial-to-trial variability between naïve and expert mice in different areas.

The revised text now reads (pg. 23): *“We performed additional single-trial analysis by calculating the trial-to-trial variance for naïve and expert mice (i.e. the trial-to-trial variance within go trials of the first or last 500 trials). We found significantly higher trial-to-trial variance in expert compared to naïve mice mainly in somatosensory cortices and M1 and only during the stim period ($p < 0.05$; Wilcoxon signed-rank test). We think this difference is mainly due to the fact that expert mice increase their body movements both in amplitude and variability during the stim period (Fig. 2b), which may lead to higher trial-to-trial variability in the relevant areas.”*

Minor

(3) Results, lines 95-96: ‘they were neither rewarded nor punished when they withheld licking for the go and no-go textures (‘correct-rejections’, CR, and ‘Misses’, respectively).’ I believe CR and Misses are transposed: a correct rejection must be withholding licking for a no-go texture, and a miss withholding for a go texture.

Fixed.

(4) Discussion, starting line 487: The authors have perhaps overlooked the retinotopic biases of higher visual areas in their discussion of signaling between anterior and posterior visual areas. Might whisker contacts with the textured surface occur below the horizontal meridian, thereby evoking activity in anterior higher visual areas, but not posterior higher visual areas?

We have now addressed this issue in the discussion (pg. 28): *“It is also possible that posterior and anterior network dissociation may relate to retinotopic properties, since posterior areas are more linked with the upper visual field⁵⁴. First, we emphasize that these experiments were performed in the dark where the incoming texture was very hard to see. Second, specifically RL and PM, on average, refer to similar retinotopic positions, especially in terms of elevation⁵⁴. Third, RL is more related to a nearby visuo-tactile space, rather than a specific retinotopic position⁵⁵. Fourth, we emphasize that we find learning-related modulations across days, whereas the trial variables (i.e. position of incoming texture and illumination) were constant, making it less likely that retinotopic parameters primarily effect our results.”*

In addition, we would like to emphasize that mice performed this task in the dark (see also point 2 for reviewer #3). The incoming textures are hard to see. To record whisker and body movements we illuminated the mouse with IR light (940 nm). The only light in the setup was the blue illumination from the top which was focused through the second objective onto the wide-

field preparation. Although this light could possibly scatter from the preparation onto the textures, the illumination conditions were still very low. In addition, all mice could perform the task in complete darkness, i.e. without excitation light and IR illumination. Therefore, it seems unlikely that visual information is relevant to this task.

We have now added this to the discussion (pg. 28): *“First, we emphasize that these experiments were performed in the dark where the incoming texture was very hard to see.”*

We have also emphasized this issue in the Methods section (pg. 36): *“Importantly, mice performed this task in the dark where motor parameters were collected using infra-red light. The only light in the setup was the blue illumination pattern that was focused through the second objective onto the wide field preparation. Illumination conditions were still very low despite this light and mice could perform the task in complete darkness. Thus, it is unlikely that visual cues from the incoming textures could affect responses in different association areas.”*

Finally, we have also added *“in the dark”* to the abstract when describing the task.

(5) Methods, line 536: At what age was the intraperitoneal injection administered?

IP injection was performed 3-5 days post-surgery (i.e. at the age of 1-4 months). This information has been added to the text.

(6) Methods, line ~557: The authors should provide some information on the lick detector (image-based? IR beam break? capacitive? piezo?). Do they have estimates of the failed detection and false detection rates?

In this study, we used a piezo sensor to detect licking events. This sensor was used in numerous previous studies and was found to be highly reliable. When observing an example body camera video, we did not observe any false positives (i.e. the sensor detected a false lick) or negatives (i.e. the sensor did not detect a lick).

The Methods section now reads (pg. 32): *“A water reward (~3 μ L) was given to the mouse for licking for the go texture only after the response cue (‘hit’), i.e. for the first correct lick during the response period (Fig. 1e; licks were detected using a piezo sensor).”*

(7) Supplementary figure 1 legend needs revision. The description of a schematic in panel a needs to be deleted. Panel b describes maps ‘for two example mice’ when maps are provided for all 7 mice.

Fixed.

Reviewer #3 (Remarks to the Author):

Gilad and Helmchen present a well-written and comprehensive analysis of cortex-wide changes in brain activity across the learning trajectory. Presentation of data from multiple animals with diverse learning rates and variable responses is important, as there is considerable heterogeneity of responses across different areas (for example, some animals show a strong cue response and others do not). Capturing this biological variability is an important addition to the field. The authors show that suppression of activity in RL and PM, two areas related to visual processing, can predict learning curves hundreds of trials in advance. This is potentially the most important and interesting finding, but its significance is not adequately discussed. Otherwise, the experimental design and analysis are clear and convincing and carried out to a very high standard. This paper should thus be an important addition to the field.

Thank you for your comments. Please see below our point-to-point response:

1. It appears that BC activity is enhanced prior to the stimulus phase, specifically in expert animals. What is the cue for this?

Indeed, the rise in BC seems to start after the initial auditory cue that indicates the trial start.

This result is not adequately emphasized or discussed.

Please see below our addition to this point in the revised MS.

Also, how many trials are averaged for the naïve animals?

Similar to the expert mouse – 500 trials.

It still seems like there is some pre-stim activity in BC before the stim arrives – this may be an early indication of expectation or plasticity in the circuit. When exactly is the BC activity initiated?

Please see analysis below in the newly added Supplementary Fig. 5.

How does this timing change with learning?

Please see below. Expert mice display earlier enhancement in BC.

Can the naïve period be restricted to fewer trials, at the very earliest stages of presentation?

Are the mice preconditioned in some way?

Yes, mice are preconditioned for one day to lick for reward after ‘go’-texture (presented within a similar trial structure as the task itself; this information is now added to the Methods section; Pg. 32). Imaging was not performed during preconditioning, therefore we do not have BC responses during these very first trials when the mouse was confronted with the experimental setup. In addition, we did not find a significant difference in BC responses between the first 50 imaged trials and the first 500 imaged trials ($P > 0.05$; Signed rank test), indicating that the very early stages of stimulus presentation (given the 1-day conditioning) are not profoundly different than our definition of naïve mice (i.e. the first 500 trials).

The revised text in the Methods section now reads (pg. 37): *“Naïve and expert mice are defined as the first and last 500 trials respectively. Responses during the first 50 trials were similar to the first 500 trials ($p > 0.05$; Signed rank test across mice), indicating that the very early stages of stimulus presentation were not substantially different.”*

We have now added additional analysis in order to quantify anticipatory responses in BC. This is now presented in a new paragraph in the results (pg. 11) and a new Supplementary figure 5: “*In addition, we found that, especially in expert mice, BC displayed enhanced activity prior to the stim period, initiating just after the stimulus cue (Fig. 3b). This finding implies anticipatory activity in BC that develops during learning. To quantify anticipatory responses in BC, we first aligned BC responses to the first touch of the whiskers to the incoming texture. Alignment was done for each trial separately and we then averaged over trials to obtain a first-touch-triggered response. BC displayed an initial mild rise in pre-touch activity followed by a salient response to the texture touch (Supplementary Fig. 5a). The onset of this initial rise occurred significantly earlier in expert compared to naïve mice ($p < 0.05$; Wilcoxon signed-rank test; Supplementary Fig. 5b). This early onset could be anticipatory activity, but it could also be related to different motor parameters. To dissociate between anticipatory effects and motor parameters we aligned the body movement vector to the first touch, and compared movement onset to onset of BC activity. BC onset was significantly earlier compared to movement onset ($p < 0.05$; Wilcoxon signed-rank test; Supplementary Fig. 5c, d; Similar results were obtained using the whisking envelope), implying that mice display a plastic expectation signal in BC that develops during learning.*”

Supplementary Figure 5 | Anticipatory activity in BC is enhanced during learning. **a**, BC response aligned to the first touch of the whiskers on the incoming texture (time 0 marks texture touch) for naïve (gray) and expert (cyan) in two example mice. Onset of responses (i.e. onset of initial rise) are marked with arrows. **b**, Response onsets in BC for naïve vs expert in all 7 mice. **c**, BC response (cyan) plotted against body movement vector (black) in two example expert mice. Both curves are normalized between 0 and 1. BC response and movement onsets are marked in arrows. **d**, BC response versus the movement onsets for all expert mice.

2. The major finding presented is that some (apparently) visual-associated areas (PM and RL) are suppressed as the animals develop expertise in the task. Because the tactile stimulus has visual characteristics (it swings into place, and the animals appear to be in a lighted area so that whisker touch data can be collected), it may be that the mouse is learning to disregard the visual aspects of the stimulus (since the texture plate swings into place in both types of trials) and instead focus on the tactile aspect of the stimulus. Thus, the downregulation of activity in these areas may not be a canonical feature of sensory learning (or of tactile learning more specifically), but rather may be related to this particular behavioral paradigm which includes both visual and tactile stimuli. This should be explicitly discussed and evaluated. Because the experiments and the analysis have been rigorously implemented, and the data presented are very clear, I do not think that this necessarily requires new experiments. However, this caveat should be discussed in both the abstract and the body of the paper.

First, we would like to emphasize that mice performed this task in the dark. The incoming textures are hard to see. To record whisker and body movements we illuminated the mouse with IR light (940 nm). The only light in the setup was the blue illumination from the top which was focused through the second objective onto the wide-field preparation. Although this light could possibly scatter from the preparation onto the textures, the illumination conditions were still very low. In addition, all mice could perform the task in complete darkness, i.e. without excitation light and IR illumination. Therefore, it seems unlikely that visual cues are relevant to this task (see also point 4 in reviewer #2 commenting on retinotopic parameter differences).

We have now added this to the discussion (pg. 29): *“First, we emphasize that these experiments were performed in the dark where the incoming texture was very hard to see.”*

We have also emphasized this issue in the Methods section (pg. 36): *“Importantly, mice performed this task in the dark where motor parameters were collected using infra-red light. The only light in the setup was the blue illumination pattern that was focused through the second objective onto the wide field preparation. Illumination conditions were still very low despite this light and mice could perform the task in complete darkness. Thus, it is unlikely that visual cues from the incoming textures could affect responses in different association areas.”*

Finally, we have also added *“in the dark”* to the abstract when describing the task.

3. Please speculate what the neurophysiological mechanisms are for enhancement of the go (stimulus-rewarded) response versus the no-go response in S1. This must be due to input from some other brain area? Surely you are not picking up differences in direct sensory input that correspond to specialized neural circuits for texture discrimination?

We have now added the following sentences to the discussion regarding possible mechanisms for the pronounced reorganization (pg. 27; also in response to point 11 of reviewer #1): *“Continuing this line, in this study we focused on cortex-wide layer 2/3 excitatory neurons, but learning-related dynamics may involve other circuit elements such as deep cortical layers^{37,38}, inhibitory subtypes³⁹ or subcortical areas^{38,40}. For example, activity of layer 6 neurons was shown to decrease the gain modulation of superficial layers³⁷, which may lead to the initial global suppression in association areas. Another possibility is that higher-order thalamocortical connections may drive synaptic plasticity during learning³⁸ and may be specific to distinct association areas such as RL⁴⁰. Specifically, these factors may also contribute to the enhanced discrimination between go and no-go trials in BC for expert mice, in which a possible integration of anticipatory signals from RL and higher-order thalamocortical inputs may*

enhance the population activity in BC for ‘go’ textures. Future studies should expand our observations to other cortical layers, cell-types, and brain areas in order to gain a more comprehensive understanding of learning-related mechanisms”

How can BF activity be increased for the Go texture prior to the texture presentation?

The early activity in BC was already discussed under point 1. We therefore presume the reviewer refers here to the discrimination plot in Fig. 7b. Perhaps there is a misunderstanding here as Fig. 7a,b were not fully clear. The vertical dashed line in these panels indicates the texture stop. As explained the first touch typically occurs at slightly different times on average around -0.33 second before the texture reaches its final stop (time 0 in Fig. 7b; the labeling is now made clearer). In Fig. 7b we therefore also plotted the histogram of first touch times below the graph to illustrate that the time point when AUC discrimination becomes significant (and when the major ‘kink’ in the BC signals occurs) coincides with the peak of the first-touch time histogram. As validation, we now in addition calculated the AUC for the BC responses aligned to the first touch, showing that AUC significantly increase only after the first touch. We have now added this analysis to Supplementary Fig. 5 also quantifying the discrimination in BC before and after the first touch (panels e and f).

Supplementary Figure 5 | Anticipatory activity in BC is enhanced during learning. e, ROC-AUC values for go vs. no-go trials for BC responses aligned on the first touch (time 0). Examples from two mice (for m1 the plot is equivalent to Figure 6b where BC responses are not aligned to first touch). Dashed gray lines indicate mean \pm 2 s.d. of shuffled data. **f,** AUC values averaged across all mice during pre and post touch (gray bars in e). Error bars are s.e.m. across mice. * $p < 0.05$; n.s. – not significant; Wilcoxon signed-rank test.

The revised text now reads (pg. 22): “Discrimination power starts to increase shortly after the first touch of the whiskers on the texture (several hundred of milliseconds before texture stop; Supplementary Fig 5e,b displays discrimination power aligned to first touch).”

Are the mice getting some auditory cue about what the texture will be, as the rotor moves into place?

No, the auditory initial trial-start cue was identical for all trial types. The increase in discrimination power is explained in the point above.

The revised text in the methods section now reads (pg. 32): “*The stimulus and response cues were identical in both textures.*”

4. By pairwise correlation, do decreased RD and increased RL activity provide better predictors of the inflection point than BC and RM changes (this is what appears to be the case in Figure 4a)?

We have reported in pg. 15: *“In addition, for each of these four areas the inflection point positively correlated with the behavioral learning threshold across mice ($r = 0.97, 0.97, 0.79$ and 0.79 for BC, RL, PM and RD respectively; $p < 0.05$).”* These correlation coefficient values mean that RL and BC are better predictors than RD and PM.

If you just use those two features, how good is the inflection point prediction?

We are not exactly sure what the reviewer means. The prediction was done for each area separately, i.e. we correlated the inflection point for each area with the learning threshold across mice.

Why does Figure 4B plot “steepest point” and not some minimum $\Delta F/F$, since that seems to be more directly related to transitions in both performance and also activity in BC and RL?

The steepest point measure was taken since it is the most intuitive for quantifying activity changes. The maximum change in a cortical area can be easily related to the maximum change in performance, i.e. the point of learning. We note that on average the steepest point in both BC and RL did not significantly differ from the learning threshold across mice (Fig. 4c). This means that some areas have a typical unidirectional change in activity in which the steepest point corresponds to the learning threshold of the mouse. Nevertheless, we did perform additional analysis where we applied a two-phase model to all areas by fitting the learning-related $\Delta F/F$ signals in cue-, pre- and stim-period with a double sigmoid (Supplementary Fig. 9). In this analysis one can derive a more optimal prediction point for each area and each time period, where in some cases the minimal point of $\Delta F/F$ may be more suitable to predict the learning threshold (e.g. Supplementary Fig. 8).

The revised text now reads (pg. 17): *“In some cases, it is also shown that a good prediction of the learning threshold is the minimal $\Delta F/F$ values at the turning point between the sigmoids.”*

It’s almost as if RD has a net inhibitory influence over RL – what is known about this circuitry?

We do not believe this is completely true, since the initial suppression is observed in most association areas, including RL (Fig. 3d and Supplementary Fig. 9). Therefore, at the initial pre-learning phase RL and RD co-vary relatively together (i.e. suppressing their activity together). This implies that RD does not necessarily have a net inhibitory influence on RL, but rather that these areas may have a common inhibitory influence which may be later dissociated during learning by a specific area targeting RL (see also discussion in point 3 above). In terms of the literature, there is a projection from RD to RL, and this connection is thought to be part of the network involved in spatial memory and navigation. It may be that in this task (in which spatial navigation is not needed) RD to RL projections are suppressed in the initial pre-learning phase.

The revised text in the discussion now reads (pg. 26): *“Suppression was also found in other association areas, indicating a possible inhibitory control mechanism arriving from a common source area. In our interpretation, suppression during the cue-period may indicate a general attentive state that is a prerequisite for learning.”*

We also add here the part in the discussion regarding possible roles for RD and RL in relation to the specific task (unchanged from the previous version; pg. 28): *“RD is also connected with*

hippocampal regions and has been shown to convey top-down effects in a visual discrimination task⁴ and has been linked to spatial navigation and memory⁵¹⁻⁵³. Therefore, it may be that under our experimental conditions, where spatial navigation is not relevant, this network is actively suppressed. Future studies may investigate whether in tasks distinct from ours, where spatial aspects are important, posterior association areas may show enhanced activity while anterior association areas including RL may be suppressed. Alternatively, RL could be the association area for processing tactile information whereas PM serves as association area for visual information and is not needed in our task.”

5. The discussion about why PM and RL are suppressed is not adequate. Especially because this is a main finding, the authors are invited to speculate more about the significance of the observation, how it is related to prior studies in NHPs, and neural structures that might be critical to generate this suppression (they mention that there is inhibition – but, is that typical? That there is inhibitory input from one area to the next?).

We have now extended the discussion relating to the suppression in PM and RD (also related to point 11 of reviewer #1 and point 1 of reviewer #2; pg. 27): *“Continuing this line, in this study we focused on cortex-wide layer 2/3 excitatory neurons, but learning-related dynamics may involve other circuit elements such as deep cortical layers^{37,38}, inhibitory subtypes³⁹ or subcortical areas^{38,40}. For example, activity of layer 6 neurons was shown to decrease the gain modulation of superficial layers³⁷, which may lead to the initial global suppression in association areas. Another possibility is that higher-order thalamocortical connections may drive synaptic plasticity during learning³⁸ and may be specific to distinct association areas such as RL⁴⁰.”*

Also on page 28: *“As for the initial suppression in PM and RD, we speculate that activating these areas (resulting in a similar activation across association areas) during pre-learning, may prolong the learning curve of the mouse. In contrast, creating a salient activation pattern across association areas (e.g. enhancing RL while suppressing PM and RD) in naïve mice, may facilitate learning. Future studies will focus on the effects of different association areas during pre-learning periods.”*

And on page 28: *“This suppression can be mediated through long-range inputs from other areas targeting inhibitory cell types⁵⁴⁻⁵⁶ or from long-range GABAergic cell originating from subcortical areas such as the hippocampus⁵⁷⁻⁵⁹.”*

In summary, the discussion has been expanded to include additional interpretations of the results in light of the reviewers’ comments. We feel that this is a positive addition to the manuscript and thank the reviewers for their suggestions.

Minor

1. There is a considerable amount of jargon in the manuscript that impedes readability. Cf “inflection point” “learning map” “Pre-period” In Figure 3c, the red line (in the deltaF/F histograms) indicates the first whisker touch in a given trial. What is the dashed cyan line for? It may be misleading to suggest it is the stim, as the stim must be present for first touch to occur. If it is the center of the stimulus presentation time, it is confusing for the reader. In any case, it should be defined in each legend where it is used.

We tried to either use standard terminology or to come up with concise, clearly defined terms for important features.

- ‘Inflection point’ is the standard term for a sigmoidal function.

- we clearly introduced and defined the three analysis windows (cue-period, pre-period, and stim-period) on pg. 7. We believe that the definition of these concise terms and their consistent use throughout the manuscript actually improves the readability.

- we also aimed to have clear and descriptive terms for the different types of maps, e.g. ‘learning map’. These terms are clearly defined at first occurrence, e.g. on pg. 17 for the learning map.

As for figure 3c, the dashed cyan line is the time of texture stop. That is, the texture approached the whisker pad at a constant speed and the mouse was able to whisk upon it as it approached. That is why we mark both the texture stop (dashed cyan) and first touch (red line). After this the textures stayed in place for 2 additional seconds. The Figure 3c legend now reads: *“Red lines indicate mean first touch of the whiskers on the incoming texture. Dashed cyan line indicates texture stop. Dashed red line indicates learning threshold.”*

2. Figure 4d, e might be better as a Supplemental Figure, as the changes in A1 activity are interesting but rather unexplained.

We feel that A1 exemplifies the observed variability across animals (some have a net increase while others show a net decrease) and the variability across learning (decreasing and then increasing activity). This variability is a central part of this study and we feel this should be kept in the main text. It also motivated the double-sigmoid fitting analysis, which is excluded in the main text and displayed in Supplementary Fig. 9.

3. In Figure 5, it appears that many (most?) brain areas are correlated with learning in the stim period. The special focus on BF is hard to interpret given this widespread effect.

This is correct. We write about the general high correlation in many other areas other than BC (*“Other sensory and motor areas showed strong correlation, too, presumably reflecting behavior-related neural activity”*). Despite this, on average (Fig. 5d), BC still shows the highest correlation during the stim period. Taken together with the fact that BC displays the highest discrimination power (Fig. 7e), this indicates to us that BC plays an important role in this task. Nevertheless, we added additional emphasis on the fact that during the stim period many other areas show learning-related modulation, which partly relates to changes in motor parameters (Fig. 2).

We now emphasize this in the revised text (pg 17): *“Finally, during the stim period many sensory and motor areas showed strong correlation, including BC, presumably reflecting motor-related neural activity.”*

And also on pg. 23: *“Nevertheless, during the stim period, with possible relation to changes in motor parameters (Fig. 2), many areas in somatosensory and motor cortex display high discrimination, indicating the large extent of learning-related modulations.”*

4. Figure 5a is very complicated, and this analysis seems critically dependent upon registration of a given pixel between imaging sessions. What is the spatial extent of a given pixel and how closely can they be aligned across days?

Pixel resolution was 512x512 resulting in a spatial resolution of approximately ~20 $\mu\text{m}/\text{pixel}$. During data analysis we downsampled the data to 256x256 pixels, resulting in a spatial resolution of ~40 $\mu\text{m}/\text{pixel}$. We note that registration of a single mouse across days was highly reliable. First, the position of the mouse in the imaging setup was kept similar across the consecutive imaging days. For each day we took a green reference image of the blood vessels and skull coordinates. From these coordinates it was relatively easy to align all imaging days onto one reference day. When comparing all the aligned reference images, we could not observe any spatial shifts between days. Therefore, the same pixel could be reliably tracked across days. As an alternative method we calculated the correlation between the learning curve and the average activity of each area (where we averaged across pixels, thus minimizing possible small shifts of single pixels). Results were similar in both cases.

5. For Figure 6f, what is the justification for separating anterior and posterior association areas?

The justification for this separation is partly based on anatomy where posterior areas are more connected within each other and to parahippocampal and hippocampal region. Anterior areas are more connected to frontal motor cortex. It is also partly based on our functional observations in the seed pixel analysis (Fig. 6b), where we observed a dissociation between anterior and posterior association areas.

The revised text now reads (pg. 36): *“For further quantification, and based on both anatomical projections^{29,32} and the functional observations from the seed maps (Fig. 6b), we divided the association cortex into anterior (RL, A, AM and AL) and posterior (PM, RD, LM, LI, PL, PR and RA) areas (see dashed red line in Figure 6c).”*

6. On p13, please define “inflection point” for the normalized curves here. This is a key point and the significance of the observations are lost in the jargon.

“Inflection point” is not jargon but is the standard mathematical term for the point of maximal steepness of a sigmoidal curve. We now clarified this definition by writing:

“The inflection point (which is the point of maximal steepness that occurs for a normalized curve where it crosses 0.5).”

7. Please speculate what the neurophysiological mechanisms are for enhancement of the go (stimulus-rewarded) response versus the no-go response in S1. This must be due to input from some other brain area? Surely you are not picking up differences in direct sensory input that correspond to specialized neural circuits for texture discrimination? How can BF activity be increased for the Go texture prior to the texture presentation? Are the mice getting some auditory cue about what the texture will be, as the rotor moves into place?

This point is identical to point 3 above

8. In Figure 8a, why is “phase 2” (adjacent to “naïve”) above “phase 1” on the y-axis of the schematic?

Fixed.

9. What is “obsessive licking” (line 478-9)?

This sentence has now been omitted.

10. Line 96, add 'Fig. 1e' to help readers find the task design.

Done.

11. Figure 1e: Can you describe exactly when water was delivered and for how long?

A small water drop (~3 μ L) was delivered only for the 'go' texture after the first lick during the reward period.

The revised text now reads (pg. 32): *"A water reward (~3 μ L) was given to the mouse for licking for the go texture only after the response cue ('hit'), i.e. for the first correct lick during the response period (Fig. 1e; licks were detected using a piezo sensor)"*

12. Line 219, Supplementary Fig. 5: Is this the right figure? Supplementary Fig. 5 is across many trials over days, whereas the sentence seems to be talking about sequential activity within a trial.

This was a mistake. We have now added the right Supplementary Fig. 4 showing the time course for A1, RL and BC in all mice.

13. Fig. 3b: Can we see the same graph averaged over all animals? Also including RD and PM?

Done. The grand average of all 5 areas is now added in Supplementary Fig. 4b.

14. Fig 3b: Please add descriptions about the black histogram inset in the legends.

Done. *"Histograms at the bottom indicate distribution of the first touch of the whiskers on the texture."*

15. Fig. 3d, pre: Why LI is highlighted in light purple instead of PM? (This appears to be a mistake)

This was a mistake. Fixed.

16. Methods: Why do you use triple transgenic mice if you don't actually need or use the tet-off system?

The triple transgenic mouse specifically labels with GCaMP6f only excitatory L2/3 neurons. This is enabled because of the triple transgenic line where the Rasgrf2-2A-dCre line targets layer 2/3, the CamK2a-Tta line targets excitatory neurons and TITL-GCaMP6f is the calcium indicator. The tet-off system in our case is redundant.

To our knowledge there was no other transgenic line with specific L2/3 neuronal expression of GCaMP6f available at the time. Only recently the Allen Institute introduced new TIGRE lines, which would allow to use double transgenic lines for similar purposes (Daigle, T. L., ..., Zeng, H. Cell, 2018).

17. Fig 8: Are phase 1 and phase 2 switched in the y-axis?

Yes. This was a mistake and is fixed.

18. Fig 8b: Is this about physical movement response of mice? I think instead of this, a schematic graph similar to Fig 3b including the key regions (A1, RD, PM, RL, BC) would be a good visualization of the key findings.

We would like to clarify that Fig. 8b is not about movement response of mice. It summarizes the activity in association areas during the pre-period (i.e. in preparation of the incoming texture). During the naïve case, activity is rather homogeneous across association areas, whereas in the expert case task-relevant area are enhanced and task-irrelevant areas are suppressed. We have now modified the y-axis in the panel and revised the figure legend.

Fig. 8 | Learning starts with a general suppression phase followed by a specific enhancement phase. a, A schematic illustration of the main cortical changes within the two temporal scales: trial (x-axis) and learning (y-axis). Two phases occur across learning: before mice actually learn to discriminate between textures, association areas display non-specific suppression of activity in response to the cue (phase 1: non-specific suppression). Only later, as mice learn the task, a specific sequential pattern is enhanced: starting from A1 in response to the cue, then RL just before the texture touches the whiskers, and BC during texture touch (phase 2: specific enhancement). Association areas PM and RD specifically remain suppressed. **b,** Schematic diagram showing the distribution of activity across association areas during the pre-period (i.e. in preparation of the upcoming texture touch) for the naïve (gray) and expert (black).

19. Methods: Please add more description about TMP injections. How many injections per mice and exactly when?

Done. The revised text now reads (pg. 31): “For TMP induction, mice were given a single intraperitoneal injection (150 μ g TMP/g body weight; 29g needle; 3-5 days post-surgery), diluted in 0.9% saline solution.”

Reviewers' Comments:

Reviewer #1:

Remarks to the Author:

The authors have satisfactorily addressed my concerns and improved the readability of their manuscript.

Reviewer #2:

Remarks to the Author:

I am entirely satisfied with the authors' answers to my questions. This is an excellent paper.

Reviewer #3:

Remarks to the Author:

I am satisfied with the authors' revisions.

Response to reviewers:

Manuscript # NCOMMS-19-26125-T

**Spatiotemporal refinement of signal flow through association cortex during learning
by Ariel Gilad and Fritjof Helmchen**

Reviewer #1 (Remarks to the Author):

The authors have satisfactorily addressed my concerns and improved the readability of their manuscript.

Reviewer #2 (Remarks to the Author):

I am entirely satisfied with the authors' answers to my questions. This is an excellent paper.

Reviewer #3 (Remarks to the Author):

I am satisfied with the authors' revisions.

We thank the reviewers for their help in improving the manuscript. We had no further revisions.